# Follow-the-Perturbed-Leader Nearly Achieves Best-of-Both-Worlds for the $m$-Set Semi-Bandit Problems

**Jingxin Zhan**
School of Mathematical Sciences
Peking University
Beijing, China 100871
bjdxzjx@pku.edu.cn

**Yuchen Xin**
School of Mathematical Sciences
Peking University
Beijing, China 100871
2301110087@pku.edu.cn

**Chenjie Sun**
School of Mathematical Sciences
Peking University
Beijing, China 100871
scj233@stu.pku.edu.cn

**Zhihua Zhang**
School of Mathematical Sciences
Peking University
Center for Intelligent Computing
Great Bay University, China
zhzhang@math.pku.edu.cn

## Abstract

We consider a common case of the combinatorial semi-bandit problem, the $m$-set semi-bandit, where the learner exactly selects $m$ arms from the total $d$ arms. In the adversarial setting, the best regret bound, known to be $\mathcal{O}(\sqrt{nmd})$ for time horizon $n$, is achieved by the well-known Follow-the-Regularized-Leader (FTRL) policy. However, this requires to explicitly compute the arm-selection probabilities via optimizing problems at each time step and sample according to them. This problem can be avoided by the Follow-the-Perturbed-Leader (FTPL) policy, which simply pulls the $m$ arms that rank among the $m$ smallest (estimated) loss with random perturbation. In this paper, we show that FTPL with a Fréchet perturbation also enjoys the near optimal regret bound $\mathcal{O}(\sqrt{nm}(\sqrt{d\log(d)} + m^{5/6}))$ in the adversarial setting and approaches best-of-both-world regret bounds, i.e., achieves a logarithmic regret for the stochastic setting. Moreover, our lower bounds show that the extra factors are unavoidable with our approach; any improvement would require a fundamentally different and more challenging method.

## 1 Introduction

The combinatorial semi-bandit problem [Cesa-Bianchi and Lugosi, 2012] is an important online decision-making problem with partial information feedback, and has many practical applications such as in shortest-path problems [Gai et al., 2012], ranking [Kveton et al., 2015a], multi-task bandits [Cesa-Bianchi and Lugosi, 2012] and recommender systems [Zou et al., 2019]. The semi-bandit problem is a sequential game that involves a learner and an environment, both interacting over time. In particular, the problem setup consists of $d$ fixed arms, and at each round $t = 1, 2, \ldots$, the learner selects a combinatorial action—a subset of arms—from a predefined set $\mathcal{A} \subset \{0, 1\}^d$. Simultaneously, the environment generates a loss vector $\ell_t \in [0, 1]^d$. The learner then incurs a loss of $\langle A_t, \ell_t \rangle$, where $A_t \in \mathcal{A}$ is the selected action, and receives semi-bandit feedback $o_t = A_t \odot \ell_t$, representing the losses associated with the selected arms only (here, $\odot$ denotes element-wise multiplication).

39th Conference on Neural Information Processing Systems (NeurIPS 2025).

In this work, we focus on a common instance of the semi-bandit setting, the $m$-set semi-bandit [Kveton et al., 2014], where each action consists of exactly $m$ arms. That is, the action set is given by $\mathcal{A} = \{a \in \{0,1\}^d : \|a\|_1 = m\}$, with $1 \leq m \leq d$. The performance of the learner is quantified by the pseudo-regret, defined as $\text{Reg}_n := \mathbb{E}\left[\sum_{t=1}^n \langle A_t - a_\star, \ell_t \rangle\right]$, where $a_\star = \arg\min_{a \in \mathcal{A}} \mathbb{E}\left[\sum_{t=1}^n \langle a, \ell_t \rangle\right]$ represents the optimal fixed action in hindsight. The expectation is taken over the randomness of both the learner's decisions and the loss. The combinatorial semi-bandit problem has been studied primarily under two frameworks: the stochastic setting and the adversarial setting.

In the adversarial setting, no assumptions are made about the generation of the loss vectors $\ell_t$; they can be chosen arbitrarily, possibly in an adaptive manner [Kveton et al., 2015a, Neu, 2015, Wang and Chen, 2018]. The optimal regret bound is $\mathcal{O}(\sqrt{nmd})$ [Audibert et al., 2014] (when $m \leq d/2$). In the stochastic setting, the losses $\ell_1, \ell_2, \ldots, \ell_n \in [0,1]^d$ are independent and identically distributed samples drawn from an unknown but fixed distribution $\mathcal{D}$. For each arm $i \in \{1, \ldots, d\}$, the expected loss is denoted by $\nu_i = \mathbb{E}_{\ell \sim \mathcal{D}}[\ell_i] \in [0,1]$. The suboptimality gap of arm $i$ is expressed by $\Delta_i := (\nu_i - \max_{\mathcal{I} \subset \{1,\cdots,d\}, |\mathcal{I}| < m} \min_{j \notin \mathcal{I}} \nu_j)^+$ and the minimum gap is $\Delta = \min_{1 \leq i \leq d, \, 0 < \Delta_i} \Delta_i$. There are many algorithms that were shown to achieve logarithmic regrets. For example, Kveton et al. [2015a] and Wang and Chen [2018] derived $\mathcal{O}(\frac{(d-m)\log(n)}{\Delta})$ regrets in $m$-set semi-bandits.

In real-world scenarios, it is often unclear whether the environment follows a stochastic or adversarial pattern, making it desirable to design policies that offer regret guarantees in both settings. To address this challenge, particularly in the classical multi-armed bandit setting, a line of research has focused on Best-of-Both-Worlds (BOBW) algorithms, which aim to achieve near-optimal performance in both regimes. A pioneering contribution in this direction was made by Bubeck and Slivkins [2012], who introduced the first BOBW algorithm. More recently, the well-known Tsallis-INF algorithm was proposed by Zimmert and Seldin [2019]. In the context of combinatorial semi-bandits, related advancements have been made by Zimmert et al. [2019], Ito [2021] and Tsuchiya et al. [2023].

However, most existing BOBW algorithms are Follow-the-Regularized-Leader (FTRL) policies and require to explicitly compute the arm-selection probabilities by solving optimizing problems at each time step and sample according to it. This problem, particularly in combinatorial semi-bandits [Neu, 2015], has attracted interest and can be avoided by the Follow-the-Perturbed-Leader (FTPL) policy, which simply pulls the $m$ arms that rank among the $m$ smallest (estimated) loss with random perturbation. More precisely, the FTPL algorithm selects the action $\arg\min_{a \in \mathcal{A}} \langle \hat{L}_t - \frac{r_t}{\eta_t}, a \rangle$, where $r_{t,i}$ denotes a random perturbation drawn from a specified distribution, $\eta_t$ is the learning rate, and $\hat{L}_{t,i}$ is an estimate of the cumulative loss for arm $i$, defined as $L_{t,i} = \sum_{s=1}^{t-1} \ell_{s,i}$.

Honda et al. [2023] first proved that FTPL with Fréchet perturbations of shape parameter $\alpha = 2$ successfully achieves BOBW guarantees in the original bandit setting (i.e., when $m = 1$), which was recently generalized by Lee et al. [2024]. They analyzed general Fréchet-type tail distributions and underscored the effectiveness of the FTPL approach. Nevertheless, in $m$-set semi-bandits the arm-selection probabilities $w_{t,i} = \phi_i(\eta_t \hat{L}_t)$ are much more complicated compared to the original setting, making it harder to analyze the regret for FTPL.

## 1.1 Contribution

In this work, we show that FTPL with Fréchet perturbations achieves $\mathcal{O}(\sqrt{nm}(\sqrt{d\log(d)} + m^{5/6}))$ regret in the adversarial regime and $\mathcal{O}(\sum_{i, \Delta_i > 0} \frac{\log(n)}{\Delta_i})$ regret in the stochastic regime simultaneously. This is the first FTPL algorithm to approach the BOBW guarantee in the semi-bandit setting when $m \leq d/2$. Technically, first, we use the standard analysis framework for FTRL algorithms (originally introduced by Lattimore and Szepesvári [2020]), and extend it to cases where the convex hull of the action set lacks interior points—i.e., to $m$-set semi-bandits—thereby simplifying Honda et al. [2023]'s proof. Second, we generalize Honda et al. [2023]'s analytical techniques and handle the challenges posed by the complex structure of arm-selection probabilities in $m$-set semi-bandits. Moreover, by establishing lower bounds, we demonstrate that our current approach has been pushed to its limit—namely, the $\log(d)$ and $\frac{m^{5/6}}{d^{1/2}}$ factors cannot be removed. Any further improvement would likely require adopting a different and more challenging line of analysis.

## 1.2 Related Works

**FTPL** The FTPL algorithm was originally introduced by Gilliland [1969] in game theory and later rediscovered and formalized by Kalai and Vempala [2005]. FTPL has since gained significant attention for its computational efficiency and adaptability across various online learning scenarios, including MAB [Abernethy et al., 2015], linear bandits [McMahan and Blum, 2004], MDP bandits [Dai et al., 2022], combinatorial semi-bandits [Neu, 2015, Neu and Bartók, 2016] and Differential Privacy [Wang and Zhu, 2024]. However, in MAB, due to the complicated expression of the arm-selection probability in FTPL, it remains a open problem [Kim and Tewari, 2019] for a long time that dose there exist a perturbation achieve the optimal regret bound $\mathcal{O}(\sqrt{nd})$ in the adversarial setting, which had been already achieved by FTRL policies [Audibert and Bubeck, 2009]. Kim and Tewari [2019] conjectured that the corresponding perturbations should be of Fréchet-type tail distribution and it was shown to be true by Honda et al. [2023], Lee et al. [2024].

**BOBW** Following the influential work of Bubeck and Slivkins [2012], a broad line of research has explored BOBW algorithms across diverse online learning settings. These include, but are not limited to, MAB [Zimmert and Seldin, 2019], the problem of prediction with expert advice [de Rooij et al., 2013, Gaillard et al., 2014, Luo and Schapire, 2015], linear bandits [Ito and Takemura, 2023, Kong et al., 2023], dueling bandits [Saha and Gaillard, 2022], contextual bandits [Kuroki et al., 2024], episodic Markov decision processes [Jin et al., 2021] and especially, combinatorial semi-bandits [Wei and Luo, 2018, Zimmert et al., 2019, Ito, 2021, Tsuchiya et al., 2023].

## 2 Preliminaries

In this section, we formulate the problem and introduce the FTPL policy.

### 2.1 The Problem Setting and Notation

We consider the $m$-set combinatorial semi-bandit problem with action set $\mathcal{A} = \{a \in \{0,1\}^d : \|a\|_1 = m\}$, where each action selects a subset of $m$ arms and $d \geq 2$. At each round $t = 1, 2, \ldots$, the learner chooses an action $A_t \in \mathcal{A}$, while the environment generates a loss vector $\ell_t \in [0,1]^d$. The learner incurs loss $\langle A_t, \ell_t \rangle$ and observes semi-bandit feedback $o_t = A_t \odot \ell_t$, i.e., the losses for the chosen arms only. The goal is to minimize the pseudo-regret $\text{Reg}_n := \mathbb{E}\left[\sum_{t=1}^n \langle A_t - a_\star, \ell_t \rangle\right]$, where $a_\star \in \arg\min_{a \in \mathcal{A}} \mathbb{E}\left[\sum_{t=1}^n \langle a, \ell_t \rangle\right]$ is the optimal fixed action in hindsight. In the adversarial setting, the loss vectors $\ell_t$ may be arbitrary and adaptive. In the stochastic setting, they are i.i.d. samples from a fixed but unknown distribution $\mathcal{D}$. Let $\nu_i := \mathbb{E}_{\ell \sim \mathcal{D}}[\ell_i]$ denote the expected loss of arm $i$. Define the suboptimality gap of arm $i$ as $\Delta_i := \left(\nu_i - \max_{\mathcal{I} \subset \{1,\ldots,d\}, |\mathcal{I}| < m} \min_{j \notin \mathcal{I}} \nu_j\right)^+$ (A less formal way to put it is: the gap from the $m$-th smallest value.), and the minimum gap as $\Delta := \min_{i:\Delta_i > 0} \Delta_i$. Here $(z)^+ = z \vee 0 := \max(z, 0)$.

To analyze regret, for any $\lambda = (\lambda_1, \ldots, \lambda_d)^T \in \mathbb{R}^d$, we let $\lambda_i$ be the $\sigma_i(\lambda)$-th smallest among the $\lambda_i$ (ties are broken arbitrarily) and $\underline{\lambda}_i := \left(\lambda_i - \max_{\mathcal{I}, |\mathcal{I}| < m} \min_{j \notin \mathcal{I}} \lambda_j\right)^+$. We denote by $\mathscr{F}_t$ the filtration $\sigma(A_1, o_1, K_1, \ldots, A_t, o_t)$, and by $\mathbf{1}$ the all-one vector. Let $a \wedge b = \min(a, b)$ and $a \vee b = \max(a, b)$.

### 2.2 Fréchet distribution

We consider the Fréchet distribution with shape parameter 2 (denoted $\mathcal{F}_2$), the density and CDF of which are
$$f(x) = 2x^{-3}e^{-1/x^2}, \quad F(x) = e^{-1/x^2}, \quad x \geq 0,$$
respectively. In the following, "Fréchet" refers to this distribution without specifying the parameter. This choice is motivated by a sequence of prior studies on the FTPL algorithm in the MAB setting [Abernethy et al., 2015, Kim and Tewari, 2019, Honda et al., 2023, Lee et al., 2024].

Briefly speaking, this choice stems from an intuitive probabilistic property of the distribution: if we draw $d$ independent samples from it, the order of the expectation of the maximum is $\sqrt{d}$, which corresponds to the optimal regret bound $\sqrt{nd}$ in the MAB problem. Our subsequent analysis (Lemma E.2) shows that this property can be generalized to the case where the order of the expectation of the

---
**Algorithm 1:** FTPL wit geometric resampling for $m$-set Semi-bandits
---
**Initialization :** $\hat{L}_1 = 0$
1 **for** $t = 1, \ldots, n$ **do**
2      Sample $r_t = (r_{t,1}, \ldots, r_{t,d})$ i.i.d. from $\mathcal{F}_2$.
3      Play $A_t = \operatorname{argmin}_{a \in \mathcal{A}} \langle \hat{L}_t - r_t/\eta_t, a \rangle$.
4      Observe $o_t = A_t \odot \ell_t$.
5      **for** $i = 1, \ldots, d$ **do**
6          Set $K_{t,i} := 0$.
7          **repeat**
8              $K_{t,i} := K_{t,i} + 1$.                  `// geometric resampling`
9              Sample $r' = (r_1', \ldots, r_d')$ i.i.d. from $\mathcal{F}_2$.
10              $A_t' = \operatorname{argmin}_{a \in \mathcal{A}} \langle \hat{L}_t - r'/\eta_t, a \rangle$.
11          **until** $A_{t,i}' = 1$
12          Set $\widehat{w_{t,i}^{-1}} := K_{t,i}$, $\hat{\ell}_{t,i} = o_{t,i} \widehat{w_{t,i}^{-1}}$, and $\hat{L}_{t+1,i} := \hat{L}_{t,i} + \hat{\ell}_{t,i}$.
13      **end**
14 **end**
---

sum of the top-$m$ results is $\sqrt{md}$, which likewise suggests the effectiveness of this distribution in the $m$-set semi-bandits problem.

## 2.3 FTPL Policy

We study the Follow-The-Perturbed-Leader (FTPL) algorithm (Algorithm 1), which selects actions based on a perturbed cumulative estimated loss $\hat{L}_t = \sum_{s=1}^{t-1} \hat{\ell}_s$. At round $t$, the learner pulls the $m$ arms that rank among the $m$ smallest estimated loss with random perturbation $r_t/\eta_t$, where $r_t \in \mathbb{R}^d$ has i.i.d. components drawn from the Fréchet distribution $\mathcal{F}_2$, and $\eta_t = \mathcal{O}(t^{-1/2})$ is the learning rate. The probability of selecting arm $i$ given $\hat{L}_t$ is $w_{t,i} = \phi_i(\eta_t \hat{L}_t)$, where for $\lambda \in \mathbb{R}^d$,

$$\phi_i(\lambda) = \mathbb{P}\{r_i - \lambda_i \text{ is among the top } m \text{ largest values in } r_1 - \lambda_1, \ldots, r_d - \lambda_d\}. \tag{1}$$

Then by Lemma C.3, we have $\phi_i(\lambda) = 2V_{i,3}(\lambda)$, where

$$V_{i,N}(\lambda) := \int_{-\lambda_i}^{\infty} \frac{e^{-1/(x+\lambda_i)^2}}{(x+\lambda_i)^N} \sum_{s=0}^{m-1} \sum_{\mathcal{I} \subseteq \{1,\ldots,d\}\setminus\{i\}, |\mathcal{I}|=s} \left[ \prod_{q \in \mathcal{I}} (1 - F(x+\lambda_q)) \prod_{q \notin \mathcal{I}, q \neq i} F(x+\lambda_q) \right] dx.$$

We denote the true cumulative loss as $L_t = \sum_{s=1}^{t-1} \ell_s$. For convenience, we also denote the vector function $\phi$ as $(\phi_1, \cdots, \phi_d)$ and $w_t$ as $(w_{t,1}, \cdots, w_{t,d})$.

**Geometric Resampling** In FTRL policies, Importance Weighted (IW) estimators are commonly used, where $\hat{\ell}_{t,i} = \frac{\ell_{t,i} A_{t,i}}{w_{t,i}}$, for $i = 1, \ldots, d$. However, in FTPL algorithms, the action probabilities $w_{t,i}$ are often hard to compute directly. To address this, the geometric resampling technique [Neu and Bartók, 2016] is frequently employed. This method replaces $w_{t,i}^{-1}$ with an unbiased estimator $\widehat{w_{t,i}^{-1}}$. Specifically, after selecting action $A_t$ and observing outcome $o_t$, for each $i = 1, \ldots, d$, we repeatedly resample $r' = (r_1', \ldots, r_d')$ i.i.d. from $\mathcal{F}_2$ and compute $A_t' = \operatorname{argmin}_{a \in \mathcal{A}} \langle \hat{L}_t - r'/\eta_t, a \rangle$ until $A_{t,i}' = 1$, i.e., arm $i$ is "selected". Let $K_{t,i}$ be the number of such resamples; then by the properties of the geometric distribution, $\mathbb{E}[K_{t,i}] = \frac{1}{w_{t,i}}$, so we define $\widehat{w_{t,i}^{-1}} := K_{t,i}$. To reduce computation, we only need to compute $K_{t,i}$ for arms actually selected by $A_t$ [Honda et al., 2023]. Since $A_{t,i} = 0$ implies $\hat{\ell}_{t,i} = 0$, the remaining estimates can be omitted.

**Viewing as Mirror Descent** FTPL can be interpreted as Mirror Descent [Abernethy et al., 2015, Lattimore and Szepesvári, 2020]. For all $\lambda \in \mathbb{R}^d$, let

$$
\begin{aligned}
\Phi(\lambda) &= \mathbb{E}[\max_{a \in \mathcal{A}} \langle r + \lambda, a \rangle] \\
&= \sum_{i=1}^{d} \mathbb{E}[(r_i + \lambda_i) \cdot \mathbb{1}_{\{ r_i + \lambda_i \text{ is among the top } m \text{ largest values in } r_1 + \lambda_1, \ldots, r_d + \lambda_d \}}].
\end{aligned}
\tag{2}
$$

Then, by exchanging expectation and the derivation (or see Lemma C.1), it is clear that $\nabla \Phi(\lambda) = \phi(-\lambda)$ and $\Phi(\lambda)$ is convex. Consider the Fenchel dual of $\Phi$, $\Phi^*(u) = \sup_{x \in \mathbb{R}^d} \langle x, u \rangle - \Phi(x)$. Then FTPL can be regarded as Mirror Descent with potential $\Phi^*$, because $w_t = \phi(\eta_t \hat{L}_t) = \nabla \Phi(-\eta_t \hat{L}_t)$. However, it is worth noting that $\nabla \Phi^*(w_t) = -\eta_t \hat{L}_t$ generally does not hold in this case, because for all $t \in \mathbb{R}, \phi(\lambda + t\mathbf{1}) = \phi(\lambda)$ by its definition, and then $\nabla \Phi$ is obviously not invertible.

## 3 Main Results

In this section, we present our main theoretical results, including the regret bounds and our new analyses of the regret decomposition.

**Theorem 3.1.** *In the adversarial setting, Algorithm 1 with learning rate $\eta_t = 1/\sqrt{t}$ satisfies*

$$
\text{Reg}_n = \mathcal{O}\left( \sqrt{nm}(\sqrt{d \log(d)} + m^{5/6}) \right).
$$

The proof is given in Section 4.3. Furthermore, Appendix C.3 provides lower bounds showing that our current method (Section 4.1) is essentially tight—indicating that the $\log(d)$ and $\frac{m^{5/6}}{d^{1/2}}$ factors are inherent to our analysis. Thus, removing them would likely require fundamentally different and more sophisticated techniques.

In the stochastic setting, we assume that there are at most $m$ arms with $\Delta_i = 0$. In other words, we assume the uniqueness of the optimal action $a_\star$. This is a common assumption in BOBW problems [Zimmert and Seldin, 2019, Zimmert et al., 2019, Honda et al., 2023].

**Theorem 3.2.** *In the stochastic setting, if the optimal action is unique, then Algorithm 1 with learning rate $\eta_t = 1/\sqrt{t}$ satisfies*

$$
\text{Reg}_n = \mathcal{O}\left( \sum_{i, \Delta_i > 0} \frac{\log(n)}{\Delta_i} \right) + \mathcal{O}\left( \frac{1}{\Delta}(m^2 d \log(d) + m^{\frac{11}{3}} + md^2) \right),
$$

*where $\Delta := \min_{i, \Delta_i > 0} \Delta_i$.*

Its proof can be found in Appendix A. Therefore, FTPL with Fréchet perturbations approaches BOBW when $m \leq d/2$. In addition, similar to Zimmert and Seldin [2019], Zimmert et al. [2019], our algorithm adopts a simple time-decaying learning rate schedule $\eta_t = 1/\sqrt{t}$. Our results can be readily extended to a more general setting with $\eta_t = c/\sqrt{t}$ for any $c > 0$.

### 3.1 Regret Decomposition

We follow the standard FTRL analysis framework for FTPL, originally by Lattimore and Szepesvári [2020, Theorem 30.4], extending it to $m$-set semi-bandits where the convex hull of the action set $\mathcal{A}$ has no interior points and hence $\nabla \Phi$ and $\nabla \Phi^*$ are not inverses of each other. For convenience, in the following, let $\eta_0 = +\infty$. Then the regret can be decomposed in the following way:

**Lemma 3.3.**

$$
\text{Reg}_n \leq \underbrace{\mathbb{E}\left[ \sum_{t=1}^{n} \langle \hat{\ell}_t, \phi(\eta_t \hat{L}_t) - \phi(\eta_t \hat{L}_{t+1}) \rangle \right]}_{\text{Stability Term}} + \underbrace{\sum_{t=1}^{n} \left( \frac{1}{\eta_t} - \frac{1}{\eta_{t-1}} \right) \mathbb{E}\left[ \Phi^*(a_\star) - \Phi^*(w_t) \right]}_{\text{Penalty Term}}.
$$

Its proof is deferred in Appendix B.1. For the penalty term, we need the following result, whose proof can be found in Appendix E.1.

**Lemma 3.4.** *For all $\lambda \in \mathbb{R}^d$, let $a = \nabla\Phi(\lambda)$. Then $\Phi^*(a) = -\mathbb{E}[\langle r, A\rangle]$, where $A = \arg\max_{a \in \mathcal{A}}\langle r + \lambda, a\rangle$. Furthermore, for all $a \in \mathcal{A}$, we have $\Phi^*(a) \leq -\mathbb{E}[\langle r, a\rangle]$.*

Combining Lemmas 3.3 and 3.4, we have

$$\text{Reg}_n \leq \underbrace{\mathbb{E}\left[\sum_{t=1}^{n}\langle\hat{\ell}_t, \phi(\eta_t\hat{L}_t) - \phi(\eta_t\hat{L}_{t+1})\rangle\right]}_{\text{Stability Term}} + \underbrace{\sum_{t=1}^{n}\left(\frac{1}{\eta_t} - \frac{1}{\eta_{t-1}}\right)\mathbb{E}\left[\langle r_t, A_t - a_\star\rangle\right]}_{\text{Penalty Term}}.$$

It is worth noting that in the decomposition of Honda et al. [2023] (see Lemmas 3 and 4 therein) and of Lee et al. [2024]) (see Lemmas 7 and 8 therein), the stability term is

$$\mathbb{E}\left[\sum_{t=1}^{n}\langle\hat{\ell}_t, \phi(\eta_t\hat{L}_t) - \phi(\eta_{t+1}\hat{L}_{t+1})\rangle\right],$$

which is further decomposed into

$$\mathbb{E}\left[\sum_{t=1}^{n}\langle\hat{\ell}_t, \phi(\eta_t\hat{L}_t) - \phi(\eta_t\hat{L}_{t+1})\rangle\right]$$

(i.e., the stability term in our decomposition) and

$$\mathbb{E}\left[\sum_{t=1}^{n}\langle\hat{\ell}_t, \phi(\eta_t\hat{L}_{t+1}) - \phi(\eta_{t+1}\hat{L}_{t+1})\rangle\right].$$

Therefore, these two terms both need to be controlled in their analysis. More specifically, the second term is bounded by $\mathcal{O}(\eta_1)$ in Honda et al. [2023], while it is bounded by $\mathcal{O}\left(\log\left(\frac{\eta_1}{\eta_{n+1}}\right)\right)$ in Lee et al. [2024]. In contrast, our decomposition does not require controlling the second term. Moreover, our penalty term is almost the same as theirs. Consequently, on the one hand, we simplify their proof; on the other hand, our bound is tighter. This also further demonstrates the superiority of the FTRL-based analysis framework.

**Remark 3.1.** *Furthermore, by Generalized Pythagoras Identity (Lemma G.2), for the stability term, one can obtain a tighter upper bound $\sum_{t=1}^{n}\frac{1}{\eta_t}\mathbb{E}[D_\Phi(-\eta_t\hat{L}_{t+1}, -\eta_t\hat{L}_t)]$, which is more popular in the analyses of FTRL policies and usually approximated by the sum of $\eta_t\mathbb{E}[\|\hat{\ell}_t\|^2_{\nabla^2\Phi(-\eta_t\hat{L}_t)}]$. However, such an approximate relationship is difficult to establish in FTPL because $\nabla^2\Phi(-\eta_t\hat{L}_{t+1})$ and $\nabla^2\Phi(-\eta_t\hat{L}_t)$ may not be close enough.*

## 4 Proof Outline

This section begins with analyses for the stability term and the penalty term, followed by a proof for Theorem 3.1 and a sketch for Theorem 3.2, whose details can be found in Appendix A. Although our analysis follows the framework in Honda et al. [2023], directly applying their approach fails in the $m$-set semi-bandit setting due to the intricate structure of the arm selection probabilities.

### 4.1 Stability Term

For the stability terms, informally, we will show that

$$\mathbb{E}_{t-1}[\langle\hat{\ell}_t, \phi(\eta_t\hat{L}_t) - \phi(\eta_t\hat{L}_{t+1})\rangle] \lesssim \eta_t\sum_{i=1}^{d}\frac{-\frac{\partial}{\partial\lambda_i}\phi_i(\eta_t\hat{L}_t)}{\phi_i(\eta_t\hat{L}_t)},$$

and hence, the key component of the analysis lies in bounding the quantity $-\frac{\frac{\partial}{\partial\lambda_i}\phi_i(\lambda)}{\phi_i(\lambda)}$, which, by the definition, is upper bounded by $\frac{3V_{i,4}(\lambda)}{V_{i,3}(\lambda)}$. However, each $V_{i,N}(\lambda)$ is a sum over many terms. To

effectively bound this ratio, our strategy is to apply a union bound over all individual terms $\frac{V_{i,4}^{\mathcal{I}}(\lambda)}{V_{i,3}^{\mathcal{I}}(\lambda)}$ such that $|\mathcal{I}| < m$ and $i \notin \mathcal{I}$, where we define

$$V_{i,N}^{\mathcal{I}}(\lambda) := \int_{-\lambda_i}^{\infty} \frac{1}{(x+\lambda_i)^N} e^{-1/(x+\lambda_i)^2} \prod_{q \in \mathcal{I}} (1 - F(x+\lambda_q)) \prod_{q \notin \mathcal{I}, \, q \neq i} F(x+\lambda_q) \, dx. \quad (3)$$

To this end, we require the following generalization of Honda et al. [2023]'s result, whose proof can be found in Appendix C.1.

**Lemma 4.1.** *For any $\mathcal{I} \subseteq \{1, \cdots, d\}$, $i \notin \mathcal{I}$, $\lambda \in \mathbb{R}^d$ such that $\lambda_i \geq 0$ and any $N \geq 3$, let*

$$J_{i,N,\mathcal{I}}(\lambda) := \int_0^{\infty} \frac{1}{(x+\lambda_i)^N} \prod_{q \in \mathcal{I}} (1 - F(x+\lambda_q)) \prod_{q \notin \mathcal{I}} F(x+\lambda_q) \, dx.$$

*Then for all $k > 0$, $\frac{J_{i,N+k,\mathcal{I}}(\lambda)}{J_{i,N,\mathcal{I}}(\lambda)}$ is increasing on $\lambda_q \geq 0$ for $q \notin \mathcal{I}$.*

In the MAB setting (i.e., the case $m = 1$), there are no $1 - F$ terms, and we can leverage monotonicity to let certain $\lambda_q$ tend to infinity, making the corresponding $F$ terms approach 1, which greatly simplifies the form of the ratio. However, for general $m$, removing $1 - F$ terms would require sending the corresponding $\lambda_q$ to negative infinity. This is not feasible, as the monotonicity does not hold generally. To address this difficulty, we rely on the following result:

**Lemma 4.2.** *For all $\mu \geq 0$, $K \geq 1$, $N \geq 3$ and $M \geq 1$, let $\mu_i \in \mathbb{R}$ for all $1 \leq i \leq M$, and define*

$$H_N = \int_0^{+\infty} (x+\mu)^{-N} e^{-\frac{K}{(x+\mu)^2}} \prod_{i=1}^M (1 - F(x+\mu_i)) \, dx.$$

*For all $k \in \mathcal{N}^+$, we have*

$$\frac{H_{N+k}}{H_N} \leq C_{N,k} \left( \left( \frac{M}{K} \right)^{k/3} \wedge \mu^{-k} \right),$$

*where $C_{N,k}$ is a positive constant only depending on $N$ and $k$. Furthermore, if $K \geq M$, then we have*

$$\frac{H_4}{H_3} \leq C \left( \left( \frac{M}{K} \log \left( \frac{K}{M} + 1 \right) \right)^{1/2} \wedge \mu^{-1} \right),$$

*where $C$ is a positive constant.*

The proof of this lemma is tedious; it constitutes the most difficult part of the stability term analysis. Therefore, we defer it to Appendix C.2. We also have lower bounds in Appendix C.3, showing that the logarithmic term and $M^{k/3}$ are inevitable. Based on Lemma 4.2, we have the following result, whose proof is deferred in Appendix D.1.

**Lemma 4.3.** *There exists $C > 0$ such that for all $t \geq 1$ and $1 \leq i \leq d$,*

$$\mathbb{E}\left[ \hat{\ell}_{t,i} \left( \phi_i(\eta_t \hat{L}_t) - \phi_i(\eta_t \hat{L}_{t+1}) \right) \mid \mathscr{F}_{t-1} \right] \leq C \cdot \hat{\underline{L}}_{t,i}^{-1} \wedge \eta_t \begin{cases} \sqrt{\frac{m \log(d)}{\sigma_i(\hat{L}_t) - m}} & \sigma_i(\hat{L}_t) > 2m \\ m^{1/3} & \sigma_i(\hat{L}_t) \leq 2m. \end{cases}$$

Note that by the definition in Section 2.1, $\hat{\underline{L}}_{t,i}^{-1} = +\infty$ when $\sigma_i(\hat{L}_t) \leq m$. As a direct corollary, we have:

**Lemma 4.4.** *There exists $C > 0$ such that for all $t \geq 1$,*

$$\mathbb{E}\left[ \langle \hat{\ell}_t, \phi(\eta_t \hat{L}_t) - \phi(\eta_t \hat{L}_{t+1}) \rangle \mid \mathscr{F}_{t-1} \right] \leq C \eta_t (\sqrt{md \log(d)} + m^{4/3}).$$

*Proof.* By Lemma 4.3, the left-hand side is less than

$$C \eta_t \left( 2m^{4/3} + \sum_{i, \sigma_i(\hat{L}_t) > 2m} \sqrt{\frac{m \log(d)}{\sigma_i(\hat{L}_t) - m}} \right) \leq C' \eta_t (\sqrt{md \log(d)} + m^{4/3}).$$

$\square$

Finally, we also need a different upper bound making use of $\hat{L}_t$ in the stochastic environment and the proof can be found in Appendix D.2, which used a new technique compared to Honda et al. [2023]. Their proof relies on the uniqueness of the optimal arm, while there are $m$ in the $m$-set semi-bandits.

**Lemma 4.5.** *If* $\sum_{i=m+1}^{d} (\eta_t \hat{\underline{L}}_{t,i})^{-2} < \frac{1}{2m}$, *then*

$$\mathbb{E}\left[\langle \hat{\ell}_t, \phi(\eta_t \hat{L}_t) - \phi(\eta_t \hat{L}_{t+1})\rangle \mid \mathscr{F}_{t-1}\right] \le C \sum_{i=m+1}^{d} \left(\hat{\underline{L}}_{t,i}^{-1} + \eta_t d w_{t,i}\right) + m 2^{-\frac{1}{2\eta_t d}},$$

*where $C$ is an absolute positive constant.*

## 4.2 Penalty Term

Then we present our analyses for the penalty term.

**Lemma 4.6.** *For all $\lambda \in \mathbb{R}^d$, we have*

$$\Phi^*(a) - \Phi^*(\phi(\lambda)) \le 5\sqrt{md}.$$

*Furthermore, if $a = \arg\min_{a' \in \mathcal{A}}\langle a', \lambda\rangle$, then*

$$\Phi^*(a) - \Phi^*(\phi(\lambda)) \le 2 \sum_{1 \le i \le d, \sigma_i(\lambda) > m} \underline{\lambda}_i^{-1}.$$

The proof is given in Appendix E.2. It is worth noting that the first part of the result stems from a key observation: if one draws $d$ i.i.d. samples from the Fréchet distribution, then the expected sum of the top $m$ largest values among them can be upper bounded by $\mathcal{O}(\sqrt{md})$. Then clearly, we have:

**Lemma 4.7.** *For all $t \ge 1$, we have*

$$\mathbb{E}\left[\Phi^*(a_\star) - \Phi^*(w_t)\right] \le 5\sqrt{md}.$$

*Furthermore, if $\max_{1 \le i \le m} \hat{L}_{t,i} \le \min_{m+1 \le i \le d} \hat{L}_{t,i}$ and $a_\star = (\underbrace{1, \cdots, 1}_{m \text{ of } 1}, \underbrace{0, \cdots, 0}_{d-m \text{ of } 0})$, then*

$$\mathbb{E}\left[\Phi^*(a_\star) - \Phi^*(w_t) \mid \mathscr{F}_{t-1}\right] \le 2\eta_t^{-1} \sum_{i=m+1}^{d} \hat{\underline{L}}_{t,i}^{-1}.$$

## 4.3 Proof for Theorem 3.1

Combining Lemmas 3.3, 4.4 and 4.7 with $\eta_t = 1/\sqrt{t}$, we have

$$\mathrm{Reg}_n \le C \sum_{t=1}^{n} \eta_t(\sqrt{md\log(d)} + m^{4/3}) + 5 \sum_{t=1}^{n} \left(\frac{1}{\eta_t} - \frac{1}{\eta_{t-1}}\right)\sqrt{md} \le C'\sqrt{nm}(\sqrt{d\log(d)} + m^{5/6}),$$

where we applied Lemma G.5.

## 4.4 Proof Sketch for Theorem 3.2

W.L.O.G., we assume that $\nu_1 \le \nu_2 \le \cdots \le \nu_d$ and then $a_\star = (\underbrace{1, \cdots, 1}_{m \text{ of } 1}, \underbrace{0, \cdots, 0}_{d-m \text{ of } 0})$. We apply the technique by Honda et al. [2023] and hence define the event $A_t = \{\sum_{i=m+1}^{d} (\eta_t \hat{\underline{L}}_{t,i})^{-2} < \frac{1}{2m}\}$. On one hand, by Lemma 4.4, 4.5 and 4.7, one can show that

$$\mathrm{Reg}_n \le \underbrace{\mathcal{O}\left(\sum_{t=1}^{n} \mathbb{E}\left[\mathbb{1}_{\{A_t\}} \cdot \sum_{i=m+1}^{d} \hat{\underline{L}}_{t,i}^{-1} + \mathbb{1}_{\{A_t^c\}}\sqrt{\frac{m}{t}}(\sqrt{d\log(d)} + m^{5/6})\right]\right)}_{I}$$

$$+ \underbrace{\mathcal{O}\left(\sum_{t=1}^{n} \mathbb{E}\left[\sum_{i=m+1}^{d} \frac{dw_{t,i}}{\sqrt{t}}\right]\right)}_{II} + \mathcal{O}\left(md^2\right).$$

On the other hand, using the fact that

$$\operatorname{Reg}_n \geq \sum_{t=1}^n \mathbb{E}\left[\sum_{i=m+1}^d \Delta_i w_{t,i}\right] \geq \underbrace{\Delta \sum_{t=1}^n \mathbb{E}\left[\sum_{i=m+1}^d w_{t,i}\right]}_{IV},$$

in Appendix A we will show that

$$\operatorname{Reg}_n \geq \underbrace{\Omega\left(\sum_{t=1}^n \mathbb{E}\left[\mathbb{1}_{\{A_t\}} \cdot t \sum_{i=m+1}^d \Delta_i \hat{\underline{L}}_{t,i}^{-2} + \mathbb{1}_{\{A_t^c\}} \cdot \frac{\Delta}{m}\right]\right)}_{III}.$$

Hence, with $\operatorname{Reg}_n = 3\operatorname{Reg}_n - 2\operatorname{Reg}_n \leq (3I - III) + (3II - IV) + \mathcal{O}\left(md^2\right)$, one can get the logarithmic result by noting that $\hat{\underline{L}}_{t,i}^{-1} - t\Delta_i\hat{\underline{L}}_{t,i}^{-2} = \mathcal{O}(\frac{1}{t\Delta_i})$ and $\sqrt{\frac{m}{t}}(\sqrt{d\log(d)} + m^{5/6})$ and $\frac{dw_{t,i}}{\sqrt{t}}$ are less than $\frac{\Delta}{m}$ and $\Delta w_{t,i}$ respectively when $t$ is large enough. Details can be found in Appendix A.

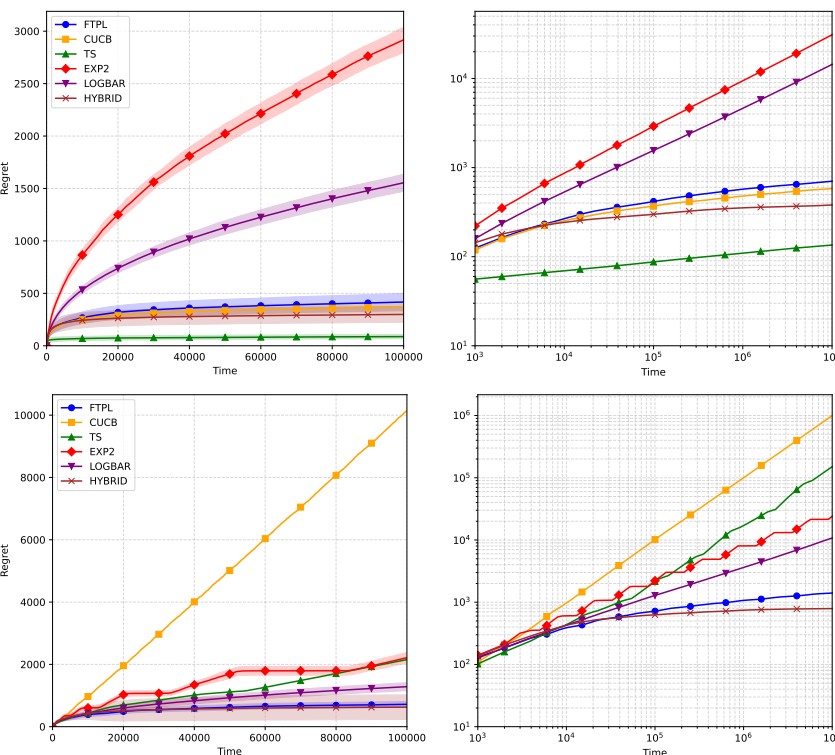

Figure 1: Comparisons of our algorithm FTPL and several existing algorithms. The left side is in linear scale and the right is in log-log scale.

## 5  Experiments

In this section, we evaluate the empirical performance of FTPL and several benchmark algorithms on the $m$-set semi-bandit problem. We compare our method against five established baselines: for the stochastic setting, we include COMBUCB [Kveton et al., 2015b] and THOMPSON SAMPLING [Gopalan et al., 2014]; for the adversarial setting, we use EXP2 [Audibert et al., 2014] and LOGBAR-RIER [Luo et al., 2018], corresponding to FTRL with generalized Shannon entropy and log-barrier regularizers, respectively. We also compare against the BOBW algorithm—FTRL with a hybrid regularizer (hereafter referred to as "hybrid") [Zimmert et al., 2019]—in both settings.

Following [Zimmert et al., 2019], we run experiments on a specific instance of the $m$-set semi-bandit with parameters $d = 10$, $m = 5$, and $n = 10^7$. The loss for arm $i$ at time $t$ has mean $\nu_{ti}$, and the realized loss is 0 with probability $1 - \nu_{ti}$ and 1 with probability $\nu_{ti}$, independently across arms and time. In the stochastic environment, the losses are generated from a stationary distribution where the mean loss for arm $i$ at time $t$ is given by $\nu_{ti} = \frac{1}{2} - \Delta$ if $i \le 5$, and $\nu_{ti} = \frac{1}{2} + \Delta$ otherwise, with $\Delta = 0.1$. In the adversarial environment, we employ the adversarial setting detailed in Zimmert et al. [2019], a framework with numerous practical applications. This setting divides the time horizon into phases: $1, \dots, n_1, n_1 + 1, \dots, n_2, \dots, n_{k-1}, \dots, n$. The duration of phase $s$ is $N_s = 1.6^s$, and the mean losses are configured as follows:

$$\nu_{ti} = \begin{cases} \frac{1}{2} - \frac{\Delta}{4} \pm \left(\frac{1}{2} - \frac{\Delta}{4}\right) & \text{if } i \le 5, \\ \frac{1}{2} + \frac{\Delta}{4} \pm \left(\frac{1}{2} - \frac{\Delta}{4}\right) & \text{otherwise,} \end{cases}$$

where $\Delta = 0.1$ and in $\pm$, $+$ is used if time $t$ falls within an odd-numbered phase, and $-$ otherwise. We sample a sequence of $n$ loss vectors from the above setting and fix it as our adversarial environment, then run the algorithms to be compared on this fixed sequence. Across all experiments, we estimated the pseudo-regret using 20 repetitions. The resulting average pseudo-regret for each algorithm over time is presented in Figure 1. Our experiments are conducted on a server with 4 NVIDIA RTX 4090 GPUs and Intel(R) Xeon(R) Gold 6132 CPU @ 2.60GHz.

## 6 Concluding Remarks

To summarize, we have shown that FTPL with Fréchet perturbations achieves both $\mathcal{O}(\sqrt{nm}(\sqrt{d \log(d)} + m^{5/6}))$ regret in the adversarial regime and $\mathcal{O}(\sum_{i, \Delta_i > 0} \frac{\log(n)}{\Delta_i})$ regret in the stochastic regime. This makes it the first FTPL algorithm to approach the Best-of-Both-Worlds (BOBW) guarantee in the $m$-set semi-bandit setting when $m \le d/2$. Our analysis has been built upon the standard FTRL framework, which we extend to accommodate the lack of interior points in the convex hull of the $m$-set action space. In doing so, we simplify and partially extend the arguments of Honda et al. [2023], and attempt to address the technical challenges arising from the intricate structure of arm-selection probabilities in the semi-bandit setting.

An important open question is whether a sharper upper bound on $\frac{V_{4,i}}{V_{3,i}}$ can be established to eliminate the $\log(d)$ and $\frac{m^{5/6}}{d^{1/2}}$ factors in the regret bound, thereby enabling FTPL to achieve the BOBW guarantee. Appendix C.3 suggests that Lemma 4.2 is already tight, meaning that these factors cannot be removed. Therefore, obtaining a tighter bound on $\frac{V_{4,i}}{V_{3,i}}$ is not possible through bounding the term-wise ratio; instead, one must analyze the ratio of the full summations directly, which is substantially more challenging [Chen and Honda, 2025]. Moreover, it has been shown that FTRL algorithms [Zimmert et al., 2019] can achieve the BOBW guarantee even in the regime where $m \ge d/2$. Whether there exists an FTPL algorithm capable of matching this performance remains an intriguing open problem. Promising future directions include extending our analysis to more general Fréchet distributions [Lee et al., 2024], investigating broader classes of the semi-bandit settings, or removing the assumption of the uniqueness of optimal actions (our additional experiments suggest that the assumption may be not essential).

## Acknowledgments and Disclosure of Funding

We would like to thank anonymous reviewers for pointing out an issue with Lemma 4.1 in the previous version. We resolved this by correspondingly modifying Lemma 4.2, and still proved that Follow-the-Perturbed-Leader approaches the Best-of-Both-Worlds for the $m$-set semi-bandit problem.

This work has been supported by the National Natural Science Foundation of China (No. 12271011 and No. 12350001) and the MOE Project of Key Research Institute of Humanities and Social Sciences (No.22JJD110001).

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

# Appendix

# A Proof for Theorem 3.2

W.L.O.G., we assume that $\nu_1 \le \nu_2 \le \cdots \le \nu_d$ and then $a_\star = (\underbrace{1, \cdots, 1}_{m \text{ of } 1}, \underbrace{0, \cdots, 0}_{d-m \text{ of } 0})$. Define the event

$A_t = \{\sum_{i=m+1}^{d} (\eta_t \underline{\hat{L}}_{t,i})^{-2} < \frac{1}{2m}\}$ and $w_\star^t = \mathbb{P}\{A_t = a_\star \mid \mathscr{F}_{t-1}\}$.

Our plan is to apply the self-bounding constrain technique by Honda et al. [2023], Zimmert and Seldin [2019]. We first derive the upper bound. On the one hand, if $\hat{L}_t$ satisfies $A_t$, which implies that $\max_{1 \le i \le m} \hat{L}_{t,i} \le \min_{m+1 \le i \le d} \hat{L}_{t,i}$, then combining Lemma 3.3, 4.5 and 4.7, the regret in round $t$ should be bounded by

$$C \sum_{i=m+1}^{d} \underline{\hat{L}}_{t,i}^{-1} + C \sum_{i=m+1}^{d} \frac{dw_{t,i}}{\sqrt{t}} + m2^{-\sqrt{t}/2d}, \tag{4}$$

where we used that $\eta_{t+1}^{-1} - \eta_t^{-1} = \sqrt{t+1} - \sqrt{t} \le \frac{1}{2\sqrt{t}} = \eta_t/2$ for the penalty term. On the other hand, if $\hat{L}_t$ doesn't satisfy $A_t$, similarly, by Lemma 3.3, 4.4 and 4.7, the regret in round $t$ is less than

$$C\sqrt{\frac{m}{t}}(\sqrt{d\log(d)} + m^{5/6}). \tag{5}$$

Putting Eq. (4) and Eq. (5) together, one can get

$$\text{Reg}_n \le C \sum_{t=1}^{n} \mathbb{E}\underbrace{\left[ \mathbb{1}_{\{A_t\}} \cdot \sum_{i=m+1}^{d} \underline{\hat{L}}_{t,i}^{-1} + \mathbb{1}_{\{A_t^c\}} \sqrt{\frac{m}{t}}(\sqrt{d\log(d)} + m^{5/6}) \right]}_{I} + C \sum_{t=1}^{n} \mathbb{E}\underbrace{\left[ \sum_{i=m+1}^{d} \frac{dw_{t,i}}{\sqrt{t}} \right]}_{II} + Cmd^2,$$

where we applied Lemma G.6 for the last term.

We then show the lower bound. Clearly, we have

$$\text{Reg}_n \ge \sum_{t=1}^{n} \mathbb{E}\left[ \sum_{i=m+1}^{d} \Delta_i w_{t,i} \right] \ge \sum_{t=1}^{n} \mathbb{E}\left[ \mathbb{1}_{\{A_t\}} \cdot \sum_{i=m+1}^{d} \Delta_i w_{t,i} + \mathbb{1}_{\{A_t^c\}} \cdot \Delta(1 - w_\star^t) \right], \tag{6}$$

where we applied Lemma F.3 for the second term. Then by Lemma F.1 and Lemma F.2, we have

$$\text{Reg}_n \ge C' \sum_{t=1}^{n} \mathbb{E}\underbrace{\left[ \mathbb{1}_{\{A_t\}} \cdot t \sum_{i=m+1}^{d} \Delta_i \underline{\hat{L}}_{t,i}^{-2} + \mathbb{1}_{\{A_t^c\}} \cdot \frac{\Delta}{m} \right]}_{III},$$

where $C'$ is an absolute positive constant. Besides, similar to Eq. (6), we also have

$$\text{Reg}_n \ge \Delta \sum_{t=1}^{n} \mathbb{E}\underbrace{\left[ \sum_{i=m+1}^{d} w_{t,i} \right]}_{IV}.$$

Hence,

$$\text{Reg}_n = 3\text{Reg}_n - 2\text{Reg}_n \le (3I - III) + (3II - IV) + Cmd^2.$$

For $3I - III$, it equals

$$\sum_{t=1}^{n} \mathbb{E}\left[ \mathbb{1}_{\{A_t\}} \cdot \left( 3C \sum_{i=m+1}^{d} \underline{\hat{L}}_{t,i}^{-1} - C't \sum_{i=m+1}^{d} \Delta_i \underline{\hat{L}}_{t,i}^{-2} \right) \right] + \sum_{t=1}^{n} \mathbb{E}\left[ \mathbb{1}_{\{A_t^c\}} \cdot \left( 3C\sqrt{\frac{m}{t}}(\sqrt{d\log(d)} + m^{5/6}) - C'\frac{\Delta}{m} \right) \right].$$

For the first term, since $ax - bx^2 \le a^2/4b$ for $b > 0$, then there exists $C'' > 0$ such that

$$3C \sum_{i=m+1}^{d} \underline{\hat{L}}_{t,i}^{-1} - C't \sum_{i=m+1}^{d} \Delta_i \underline{\hat{L}}_{t,i}^{-2} \le \sum_{i=m+1}^{d} \frac{C''}{t\Delta_i}.$$

Note that $3C\sqrt{\frac{m}{t}}(\sqrt{d\log(d)} + m^{5/6}) \leq C'\frac{\Delta}{m}$ after $\sqrt{t} \geq \frac{3Cm^{\frac{3}{2}}(\sqrt{d\log(d)}+m^{5/6})}{C'\Delta}$, then we have

$$3I - III \leq \sum_{t=1}^{n} \sum_{i=m+1}^{d} \frac{C''}{t\Delta_i} + \sum_{t=1}^{\frac{9C^2m^3(\sqrt{d\log(d)}+m^{5/6})^2}{C'^2\Delta^2}} 3C\sqrt{\frac{m}{t}}(\sqrt{d\log(d)} + m^{5/6}) \tag{7}$$

$$\leq \sum_{i=m+1}^{d} \frac{C_1\log(n)}{\Delta_i} + \frac{C_2 m^2(d\log(d) + m^{5/3})}{\Delta},$$

where $C_1$ and $C_2$ are absolute positive constants and we used $(x + y)^2 \leq 2(x^2 + y^2)$ for positive $x$ and $y$.

Then it suffices to bound $3II - IV$, which equals

$$\sum_{t=1}^{n} \mathbb{E}\left[\sum_{i=m+1}^{d}\left(\frac{3Cdw_{t,i}}{\sqrt{t}} - \Delta w_{t,i}\right)\right].$$

Similarly, $\frac{3Cdw_{t,i}}{\sqrt{t}} \leq \Delta w_{t,i}$ after $\sqrt{t} \geq \frac{3Cd}{\Delta}$. Hence,

$$\sum_{t=1}^{n} \mathbb{E}\left[\sum_{i=m+1}^{d}\left(\frac{3Cdw_{t,i}}{\sqrt{t}} - \Delta w_{t,i}\right)\right] \leq m \sum_{t=1}^{\frac{9C^2d^2}{\Delta^2}} \frac{3Cd}{\sqrt{t}} \leq \frac{C_3 md^2}{\Delta},$$

where $C_3$ is an absolute positive constant and we used that $\sum_{i=1}^{d} w_{t,i} = m$ by Lemma C.2. We complete the proof by putting everything together.

## B  Decomposition

In this section, we give the detailed proof for the regret decomposition.

**Lemma B.1.** *Let* $\ell_1, \cdots, \ell_n \in \mathbb{R}^d$ *and* $a_t = \phi(\eta_t L_t)$, *where* $(\eta_t)_{t=0}^{n}$ *is decreasing with* $\eta_0 = +\infty$ *and* $L_t := \sum_{s=1}^{t-1} \ell_s$. *Then for all* $a \in \mathcal{A}$,

$$\sum_{t=1}^{n}\langle a_t - a, \ell_t\rangle \leq \sum_{t=1}^{n}\langle \ell_t, \phi(\eta_t L_t) - \phi(\eta_t L_{t+1})\rangle + \sum_{t=1}^{n}\left(\frac{1}{\eta_t} - \frac{1}{\eta_{t-1}}\right)(\Phi^*(a) - \Phi^*(a_t)).$$

*Proof.* For convenience, let $\eta_{n+1} = \eta_n$ and $a_{n+1} = \phi(\eta_n L_{n+1})$. Note that $a_t = \nabla\Phi(-\eta_t L_t)$, then by Lemma G.1, $-\eta_t L_t \in \partial\Phi^*(a_t)$, which implies that $a_t \in \arg\min_{x\in\mathbb{R}^d} \Phi_t^*(x)$, where $\Phi_t^*(x) := \frac{\Phi^*(x)}{\eta_t} + \langle x, L_t\rangle$. We then have

$$\sum_{t=1}^{n}\langle a_t - a, \ell_t\rangle = \sum_{t=1}^{n}\langle a_t - a_{t+1}, \ell_t\rangle + \sum_{t=1}^{n}\langle a_{t+1}, \ell_t\rangle - \sum_{t=1}^{n}\langle a, \ell_t\rangle$$

$$= \sum_{t=1}^{n}\langle a_t - a_{t+1}, \ell_t\rangle + \sum_{t=1}^{n}\left(\Phi_{t+1}^*(a_{t+1}) - \frac{\Phi^*(a_{t+1})}{\eta_{t+1}} - \left[\Phi_t^*(a_{t+1}) - \frac{\Phi^*(a_{t+1})}{\eta_t}\right]\right)$$

$$- \sum_{t=1}^{n}\left(\Phi_{t+1}^*(a) - \frac{\Phi^*(a)}{\eta_{t+1}} - \left[\Phi_t^*(a) - \frac{\Phi^*(a)}{\eta_t}\right]\right)$$

$$= \sum_{t=1}^{n}\langle a_t - a_{t+1}, \ell_t\rangle + \sum_{t=1}^{n}(\Phi_t^*(a_t) - \Phi_t^*(a_{t+1})) + \sum_{t=1}^{n}\left(\frac{1}{\eta_t} - \frac{1}{\eta_{t-1}}\right)(\Phi^*(a) - \Phi^*(a_t))$$

$$+ \Phi_{n+1}^*(a_{n+1}) - \Phi_{n+1}^*(a).$$

Since for all $a \in \mathcal{A}$, $\Phi_{n+1}^*(a_{n+1}) \leq \Phi_{n+1}^*(a)$, we have

$$\sum_{t=1}^{n}\langle a_t - a, \ell_t\rangle \leq \sum_{t=1}^{n}(\langle a_t - a_{t+1}, \ell_t\rangle + \Phi_t^*(a_t) - \Phi_t^*(a_{t+1})) + \sum_{t=1}^{n}\left(\frac{1}{\eta_t} - \frac{1}{\eta_{t-1}}\right)(\Phi^*(a) - \Phi^*(a_t)).$$

Then by the definition,

$$\Phi_t^*(a_t) - \Phi_t^*(a_{t+1}) = -\frac{1}{\eta_t}\left(\Phi^*(a_{t+1}) - \Phi^*(a_t) - \langle a_{t+1} - a_t, -\eta_t L_t\rangle\right) = -\frac{1}{\eta_t}D_\Phi(-\eta_t L_t, -\eta_{t+1}L_{t+1}),$$

where we used Lemma G.3 by noting that $\nabla\Phi(-\eta_t L_t) = a_t$ and $\nabla\Phi(-\eta_{t+1}L_{t+1}) = a_{t+1}$. Finally, by Lemma G.2 (taking $x = -\eta_t L_{t+1}, y = -\eta_{t+1}L_{t+1}$ and $z = -\eta_t L_t$), we have

$$\langle a_t - a_{t+1}, \ell_t\rangle - \frac{1}{\eta_t}D_\Phi(-\eta_t L_t, -\eta_{t+1}L_{t+1}) \le \langle \ell_t, \nabla\Phi(-\eta_t L_t) - \nabla\Phi(-\eta_t L_{t+1})\rangle,$$

since $D_\Phi(x, y) + D_\Phi(z, x) \ge 0$. We complete the proof by putting them together. $\square$

**Remark B.1.** *The overall proof framework is based on* Lattimore and Szepesvári *[2020, Exercise 28.12], with the latter part inspired by* Zimmert and Lattimore *[2019, Lemma 3].*

### B.1 Proof for Lemma 3.3

*Proof.* Noting that $\mathbb{E}[A_t \mid \mathscr{F}_{t-1}] = \phi(\eta_t \hat{L}_t)$, we have

$$\text{Reg}_n = \mathbb{E}\left[\sum_{t=1}^n \left\langle \phi(\eta_t \hat{L}_t) - a_\star, \ell_t\right\rangle\right] = \mathbb{E}\left[\sum_{t=1}^n \left\langle \phi(\eta_t \hat{L}_t) - a_\star, \hat{\ell}_t\right\rangle\right].$$

Then it suffices to apply Lemma B.1. $\square$

## C  Important Facts

In this section, we present some important facts to be used in our analyses.

**Lemma C.1.** *For all $\lambda \in \mathbb{R}^d$, we have $\nabla\Phi(\lambda) = \phi(-\lambda)$ and $\Phi(\lambda)$ is convex over $\mathbb{R}^d$.*

*Proof.* By Eq. (2), since for all $1 \le i \le d$, $\mathbb{E}|r_i| < +\infty$, one can exchange expectation and the derivative, then we have

$$\frac{\partial}{\partial\lambda_i}\Phi(\lambda) = \mathbb{E}\left[\mathbb{1}_{\{r_i + \lambda_i \text{ is among the top } m \text{ largest values in } r_1 + \lambda_1, \cdots, r_d + \lambda_d\}}\right] = \phi_i(-\lambda),$$

because

$$\frac{\partial}{\partial\lambda_i}\mathbb{1}_{\{r_i + \lambda_i \text{ is among the top } m \text{ largest values in } r_1 + \lambda_1, \cdots, r_d + \lambda_d\}} = 0, \ a.s.$$

This shows that $\nabla\Phi(\lambda) = \phi(-\lambda)$. For convexity, it suffices to note that taking maximum and expectation keeps convexity. $\square$

**Lemma C.2.** *For all $\lambda \in \mathbb{R}^d$, we have $\sum_{i=1}^d \phi_i(\lambda) = m$.*

*Proof.* By the definition, we have

$$\sum_{i=1}^d \phi_i(\lambda) = \mathbb{E}\left[\sum_{i=1}^d \mathbb{1}_{\{r_i - \lambda_i \text{ is among the top } m \text{ largest values in } r_1 - \lambda_1, \cdots, r_d - \lambda_d\}}\right] = m.$$

$\square$

**Lemma C.3.** $\phi_i(\lambda) = 2V_{i,3}(\lambda)$

*Proof.* Because

$$\phi_i(\lambda) = \mathbb{E}_{r_i}[\mathbb{P}\{\text{there exist at most } m-1 \text{ of } r_1 - \lambda_1, \cdots, r_{i-1} - \lambda_{i-1},$$
$$r_{i+1} - \lambda_{i+1}, \cdots, r_d - \lambda_d \text{ that are larger than } x \mid r_i - \lambda_i = x\}],$$

then it suffices to note that the conditional probability inside is just

$$\sum_{s=0}^{m-1} \sum_{\mathcal{I}\subseteq\{1,\ldots,d\}\setminus\{i\}, |\mathcal{I}|=s} \left[\prod_{q\in\mathcal{I}}(1 - F(x + \lambda_q)) \prod_{q\notin\mathcal{I}, q\neq i} F(x + \lambda_q)\right].$$

$\square$

## C.1   Proof for Lemma 4.1

*Proof.* We follow the proof by Honda et al. [2023]. Let $Q(x) = h(x)(x + \lambda_i)^{-N} \prod_{q \in \mathcal{I}}(1 - F(x + \lambda_q)) \prod_{q \notin \mathcal{I}} F(x + \lambda_q)$. If $q \notin \mathcal{I}$, then

$$\frac{\partial}{\partial \lambda_q} J_{i,N,\mathcal{I}}(\lambda) = 2 \int_0^{+\infty} (x + \lambda_q)^{-3} Q(x) \, \mathrm{d}x := 2J_{i,N,\mathcal{I}}^q(\lambda).$$

Hence,

$$\frac{\partial}{\partial \lambda_q} \frac{J_{i,N+k,\mathcal{I}}(\lambda)}{J_{i,N,\mathcal{I}}(\lambda)} = 2 \cdot \frac{J_{i,N+k,\mathcal{I}}^q(\lambda) J_{i,N,\mathcal{I}}(\lambda) - J_{i,N+k,\mathcal{I}}(\lambda) J_{i,N,\mathcal{I}}^q(\lambda)}{J_{i,N,\mathcal{I}}(\lambda)^2}.$$

Note that

$$J_{i,N+k,\mathcal{I}}^q(\lambda) J_{i,N,\mathcal{I}}(\lambda) = \int \int_{x,y \geq 0} (x + \lambda_q)^{-3}(x + \lambda_i)^{-k} Q(x)Q(y) \, \mathrm{d}x \, \mathrm{d}y$$

$$= \frac{1}{2} \int \int_{x,y \geq 0} Q(x)Q(y) \left[(x + \lambda_q)^{-3}(x + \lambda_i)^{-k} + (y + \lambda_q)^{-3}(y + \lambda_i)^{-k}\right] \mathrm{d}x \, \mathrm{d}y,$$

and similarly,

$$J_{i,N+k,\mathcal{I}}(\lambda) J_{i,N,\mathcal{I}}^q(\lambda) = \frac{1}{2} \int \int_{x,y \geq 0} Q(x)Q(y) \left[(y + \lambda_q)^{-3}(x + \lambda_i)^{-k} + (x + \lambda_q)^{-3}(y + \lambda_i)^{-k}\right] \mathrm{d}x \, \mathrm{d}y,$$

then we have $J_{i,N+k,\mathcal{I}}^q(\lambda) J_{i,N,\mathcal{I}}(\lambda) - J_{i,N+k,\mathcal{I}}(\lambda) J_{i,N,\mathcal{I}}^q(\lambda) =$

$$\frac{1}{2} \int \int_{x,y \geq 0} Q(x)Q(y) \left[(x + \lambda_q)^{-3}(x + \lambda_i)^{-k} + (y + \lambda_q)^{-3}(y + \lambda_i)^{-k}\right.$$

$$\left. -(y + \lambda_q)^{-3}(x + \lambda_i)^{-k} - (x + \lambda_q)^{-3}(y + \lambda_i)^{-k}\right] \mathrm{d}x \, \mathrm{d}y$$

$$= \frac{1}{2} \int \int_{x,y \geq 0} Q(x)Q(y) \left[(x + \lambda_q)^{-3} - (y + \lambda_q)^{-3}\right] \left[(x + \lambda_i)^{-k} - (y + \lambda_i)^{-k}\right] \mathrm{d}x \, \mathrm{d}y \geq 0,$$

which implies that $\frac{J_{i,N+k,\mathcal{I}}(\lambda)}{J_{i,N,\mathcal{I}}(\lambda)}$ increases with $\lambda_q \geq 0$. $\qquad\square$

## C.2   Proof for Lemma 4.2

We divide the proof into two parts. Recall that for all $\mu \geq 0$, $K, M \geq 1$, $N \geq 3$ and $\mu_i \in \mathbb{R}$ for all $1 \leq i \leq M$, we defined

$$H_N = \int_0^{+\infty} (x + \mu)^{-N} e^{-\frac{K}{(x+\mu)^2}} \prod_{i=1}^M (1 - F(x + \mu_i)) \, \mathrm{d}x.$$

**Lemma C.4.** *For all $k \in \mathcal{N}^+$, we have*

$$\frac{H_{N+k}}{H_N} \leq C_{N,k} \left(\left(\frac{M}{K}\right)^{k/3} \wedge \mu^{-k}\right),$$

*where $C_{N,k}$ is a positive constant only depending on $N$ and $k$.*

*Proof.* The upper bound of $\mu^{-k}$ is obvious because $(x + \mu)^{-N-k} \leq \mu^{-k}(x + \mu)^{-N}$ and hence, in the following we assume that $\mu^{-1} \geq C_N' \left(\frac{M}{K}\right)^{1/3}$, where $C_N' = 8\sqrt{N-2}$. Let $u = \frac{1}{x+\mu}$ and

$$g(u) = u^{N-2} e^{-Ku^2} \prod_{i=1}^M \left(1 - F\left(\frac{1}{u} + \mu_i - \mu\right)\right) / \Lambda, u \geq 0,$$

where $\Lambda$ is a constant such that $\int_0^{\mu^{-1}} g(u) \, \mathrm{d}u = 1$. Consider a random variable $U$ with pdf $g(u)$, then clearly,

$$\frac{H_{N+k}}{H_N} = \mathbb{E}[U^k].$$

Our plan is to find some $u_0 > 0$ such that $g(u)$ decays rapidly after $u_0$, then we expect that the order of $E[U^k]$ is roughly $u_0^k$.

Let $\ell(u) = \log g(u)$ and $y_i = \frac{1}{u} + \mu_i - \mu$ for all $1 \leq i \leq M$, then when $u \geq 0$,

$$\frac{\mathrm{d}\ell}{\mathrm{d}u}(u) = \frac{N-2}{u} - 2Ku + 2\sum_{i=1}^{M} \frac{\mathbb{1}_{\{y_i \geq 0\}}}{u^2 y_i^3 \left(e^{\frac{1}{y_i^2}} - 1\right)}.$$

Then it suffices to show that when $u \geq u_0 := C_N' \left(\frac{M}{K}\right)^{1/3}/2$, $\frac{\mathrm{d}\ell}{\mathrm{d}u}(u) \leq -Ku/2$. Because note that $u_0 \geq 4K^{-1/3} \geq 4K^{-1/2}$ and $\mu^{-1} \geq 2u_0$, then by Lemma G.10, we have

$$\mathbb{E}[U^k] \leq u_0^k + \mathbb{E}[U^k \mathbb{1}_{\{\mu^{-1} \geq U \geq u_0\}}] \leq (1 + 2k!!)u_0^k.$$

Note that $\sup_{x \geq 0} \frac{1}{x^3 \left(e^{\frac{1}{x^2}} - 1\right)} < 1$, then we have

$$\frac{\mathrm{d}\ell}{\mathrm{d}u}(u) \leq \frac{N-2}{u} - 2Ku + \frac{2M}{u^2}.$$

When $u \geq u_0$, clearly, since $C_N$ is large enough, we have

$$\frac{2M}{u^2} \leq \frac{Ku}{2}.$$

Also, since $u_0 > \sqrt{\frac{N-2}{K}}$, when $u \geq u_0$, we have $\frac{N-2}{u} \leq Ku$. Therefore, when $u \geq u_0$, $\frac{\mathrm{d}\ell}{\mathrm{d}u}(u) \leq -Ku/2$, which completes our proof.

$\square$

**Lemma C.5.** *If $K \geq M$, then we have*

$$\frac{H_4}{H_3} \leq C\left(\left(\frac{M}{K}\log\left(\frac{K}{M}+1\right)\right)^{1/2} \wedge \mu^{-1}\right),$$

*where $C$ is a positive constant.*

*Proof.* We still use the definition of $g$ and $U$ in the proof of Lemma C.4 ($N = 3$ and $k = 1$). Let $u_0 = \left(\frac{M}{K}\log\left(\frac{K}{M}+1\right)\right)^{1/2}$. When $K \leq 32M$, by Lemma C.4, the result holds clearly when $C$ is large enough. Hence we then assume that $K > 32M$. Then $u_0 < 1/3$. Similarly, in the following, we also assume that $\mu^{-1} \geq C'u_0$, where $C' > 3$ is a large constant to be chosen and is not depending on $K$ and $M$.

Similarly, our plan is still to find some $u_0 > 0$ such that $g(u)$ decays rapidly after $u_0$. We will first compute the ratio $\frac{g(su_0)}{g(u_0)}$ for $s \geq 1$, and then find a suitable choice of $u_0$ such that this ratio decays at a rate comparable to the Gaussian tail.

However, the challenge is that we also need to ensure that $g(u_0)$ is not too large in order to concretely control the decay rate of $g(su_0)$. To illustrate this, we will use a simple fact: if we choose $u_0$ large enough so that $g(u)$ starts decreasing after $\frac{u_0}{3}$, then the values of $g(u)$ over the interval $\left[\frac{u_0}{3}, \frac{2u_0}{3}\right]$ should all be larger than $g(u_0)$. Since $g(u)$ is a probability density function, we then have

$$1 \geq \int_{\frac{u_0}{3}}^{\frac{2u_0}{3}} g(u)\, du \geq g(u_0) \cdot \frac{u_0}{3},$$

which naturally gives an upper bound on $g(u_0)$. For the convenience of presentation, in the following formal proof, we scaled $u_0$ to $3u_0$.

For all $s \geq t \geq 1$, we have

$$\frac{g(su_0)}{g(tu_0)} = \frac{s}{t}e^{-Ku_0^2(s^2-t^2)}\prod_{i=1}^{M} \frac{1 - F\left(\frac{1}{su_0} + \mu_i - \mu\right)}{1 - F\left(\frac{1}{tu_0} + \mu_i - \mu\right)}.$$

If $1 \leq t \leq 3$, then $tu_0 < 1$. Hence, by Lemma G.7, we have

$$\frac{g(su_0)}{g(tu_0)} \leq \frac{s}{t} e^{-Ku_0^2(s^2-t^2)} \left(\frac{8}{t^2 u_0^2}\right)^M \leq s e^{-Ku_0^2(s^2-t^2)} \left(\frac{8}{u_0^2}\right)^M. \tag{8}$$

Then on the one hand, for all $t \in [1, 2]$, we have

$$\frac{g(3u_0)}{g(tu_0)} \leq 3 e^{-5Ku_0^2} \left(\frac{8}{u_0^2}\right)^M \leq 3 e^{-5M \log\left(\frac{K}{M}+1\right)} \left(\frac{8K}{M}\right)^M,$$

where we used the definition of $u_0$ in the last inequality. Since $\frac{K}{M} > 8$, we have

$$\frac{g(3u_0)}{g(tu_0)} \leq 3 e^{-5M \log\left(\frac{K}{M}+1\right)+2M \log\left(\frac{K}{M}\right)} < 1.$$

Then since $C' \geq 2$,

$$1 \geq \int_{u_0}^{\mu^{-1}} g(u)\,\mathrm{d}u \geq \int_{u_0}^{2u_0} g(u)\,\mathrm{d}u \geq u_0 g(3u_0),$$

which implies that $g(3u_0) \leq u_0^{-1}$.

On the other hand, by Eq. (8), for all $s \geq 3$, we have

$$\frac{g(su_0)}{g(3u_0)} \leq s e^{-Ku_0^2(s^2-9)} \left(\frac{8}{u_0^2}\right)^M \leq e^{\log(s)-M \log\left(\frac{K}{M}+1\right)(s^2-9)+M \log(8)+M \log\left(\frac{K}{M}\right)}.$$

Then since $K > 32M \geq 32$, one can find $C'' > 3$ that is not depending on $K$ and $M$ and large enough (one can then pick an $C'$ larger than $C''$) such that for all $s \geq C''$, we have

$$\frac{g(su_0)}{g(3u_0)} \leq e^{-\frac{M}{2} \log\left(\frac{K}{M}+1\right)s^2}.$$

Therefore,

$$\int_{C''}^{\frac{\mu^{-1}}{u_0}} s g(su_0)\,\mathrm{d}s \leq g(3u_0) \int_{C''}^{+\infty} s e^{-M \log\left(\frac{K}{M}+1\right)s^2/2}\,\mathrm{d}s \leq u_0^{-1} \int_{C''}^{+\infty} s e^{-M \log\left(\frac{K}{M}+1\right)s^2/2}\,\mathrm{d}s$$

$$= u_0^{-1} \left(M \log\left(\frac{K}{M}+1\right)\right)^{-1} e^{-C''^2 M \log\left(\frac{K}{M}+1\right)/2} \leq u_0^{-1}.$$

Then we have

$$\mathbb{E}[U] \leq C'' u_0 + \int_{C'' u_0}^{\mu^{-1}} u g(u)\,\mathrm{d}u = C'' u_0 + u_0^2 \int_{C''}^{\frac{\mu^{-1}}{u_0}} s g(su_0)\,\mathrm{d}s \leq (C''+1) u_0.$$

$\square$

## C.3 Lower Bounds

In this section, we will prove lower bounds for Lemma 4.2, showing that the logarithmic term and $M^{1/3}$ are inevitable.

**Lemma C.6.** *For all $\mu \in \mathbb{R}$, $K \geq 1$, and $N \geq 3$, define*

$$U_N(\mu) = \int_0^{+\infty} x^{-N} e^{-\frac{K}{x^2}} (1 - F(x+\mu))\,\mathrm{d}x.$$

*Then there exists $C > 0$ such that for all $K \geq 2$, we have*

$$\sup_{\mu \in \mathbb{R}} \frac{U_4(\mu)}{U_3(\mu)} \geq C \sqrt{\frac{\log(K)}{K}}.$$

*Proof.* Let $w = e^{-\frac{K}{x^2}}$, then $x(w) = \sqrt{\frac{K}{-\log w}}$ and

$$U_N(\mu) = \frac{1}{2} K^{-\frac{N-1}{2}} \int_0^1 (-\log w)^{\frac{N-3}{2}} (1 - F(x(w) + \mu)) \, \mathrm{d}w,$$

which implies that

$$\sqrt{K} \frac{U_4(\mu)}{U_3(\mu)} = \frac{\int_0^1 (-\log w)^{\frac{1}{2}} (1 - F(x(w) + \mu)) \, \mathrm{d}w}{\int_0^1 [1 - F(x(w) + \mu)] \, \mathrm{d}w} = \mathbb{E}[(-\log(W))^{1/2}],$$

where $W$ is a random variable with p.d.f. proportional to $(1 - F(x(w) + \mu)) \mathbb{1}_{\{w \in [0,1]\}}$.

An intuitive understanding of this result is that the pdf of $W$, given by $1 - F(x(w) + \mu)$, is monotonically decreasing on $[0, 1]$ and clearly exhibits a sharp drop at $e^{-\frac{K}{\mu^2}}$: the pdf is equal to 1 before this point and then gradually decays afterward. Our calculation will show that most of the density is concentrated in the first half of the interval, and thus the expectation of $(-\log(W))^{1/2}$ is of a larger order than simply substituting $W = e^{-\frac{K}{\mu^2}}$. Thus, by choosing $\mu = -\sqrt{\frac{2K}{\log(K)}}$, the desired lower bound follows.

Clearly, for all $s \geq 0$, we have

$$\mathbb{E}[(-\log(W))^{1/2}] \geq \sqrt{s} \mathbb{P}(W \leq e^{-s}). \tag{9}$$

Let $\mu = -\sqrt{\frac{2K}{\log(K)}}$ and $s = \frac{K}{\mu^2} = \frac{\log(K)}{2}$. Then when $w \leq e^{-s}$, one can see that $x(w) + \mu \leq 0$, which implies that

$$\int_0^{e^{-s}} [1 - F(x(w) + \mu)] \, \mathrm{d}w = e^{-s} = K^{-1/2}.$$

Then it suffices to show that

$$\int_{e^{-s}}^1 [1 - F(x(w) + \mu)] \, \mathrm{d}w = e^{-s} = \mathcal{O}(K^{-1/2}), \tag{10}$$

since then we have

$$\mathbb{P}(W \leq e^{-s}) = \frac{K^{-1/2}}{K^{-1/2} + \mathcal{O}(K^{-1/2})} \geq C,$$

where $C$ is a positive constant. Combining Eq. (9) and Eq. (10) leads to the desired result.

To show Eq. (10), one should note that when $w \geq e^{-(K^{-1/4} + s^{-1/2})^{-2}}$,

$$1 - F(x(w) + \mu) = 1 - e^{-\frac{1}{(x(w) + \mu)^2}} \leq \frac{1}{(x(w) + \mu)^2} = \frac{1}{K \left( \sqrt{\frac{1}{-\log w}} - \sqrt{\frac{1}{s}} \right)^2} \leq K^{-1/2},$$

where we used that $1 - e^{-x} \leq x$ for all $x \in \mathbb{R}$. Then

$$\int_{e^{-(K^{-1/4} + s^{-1/2})^{-2}}}^1 [1 - F(x(w) + \mu)] \, \mathrm{d}w \leq K^{-1/2}.$$

Finally, since

$$e^{-(K^{-1/4} + s^{-1/2})^{-2}} - e^{-s} = e^{-s} \cdot \left( e^{s - s(K^{-1/4} s^{1/2} + 1)^{-2}} - 1 \right),$$

and noting that, by $s = \frac{\log(K)}{2}$,

$$s - s(K^{-1/4} s^{1/2} + 1)^{-2} = \frac{s(K^{-1/2} s + 2K^{-1/4} s^{1/2})}{(K^{-1/4} s^{1/2} + 1)^2} = o(1),$$

we then have

$$\int_{e^{-s}}^{e^{-(K^{-1/4} + s^{-1/2})^{-2}}} [1 - F(x(w) + \mu)] \, \mathrm{d}w \leq e^{-(K^{-1/4} + s^{-1/2})^{-2}} - e^{-s} = o(e^{-s}) = o(K^{-1/2}),$$

where we used that $e^{o(1)} - 1 = o(1)$. It suffices to put everything together. $\qquad \square$

**Lemma C.7.** *For all $\mu \in \mathbb{R}$, $M, K \geq 1$, and $N \geq 3$, define*

$$R_N(\mu) = \int_0^{+\infty} x^{-N} e^{-\frac{K}{x^2}} \left(1 - F(x+\mu)\right)^M \, \mathrm{d}x.$$

*Then there exists $C > 0$ such that for all $M \geq 2K$, we have*

$$\sup_{\mu \in \mathbb{R}} \frac{R_4(\mu)}{R_3(\mu)} \geq C \left(\frac{M}{K}\right)^{1/3}.$$

*Proof.* We still use the definition of $g$ and $U$ in the proof of Lemma C.4 ($N = 3$ and $k = 1$). Our plan is to show that that exists $\mu \in \mathbb{R}$ and corresponding $C > 0$ such that when $0 \leq u \leq u_0 := C\left(\frac{M}{K}\right)^{1/3}$, $\frac{\mathrm{d}\ell}{\mathrm{d}u}(u) \geq 0$, which implies that $g(u)$ increases with $u$ when $u \leq u_0$. Then we have

$$\mathbb{P}\left(U \leq \frac{u_0}{2}\right) = \int_0^{\frac{u_0}{2}} g(u) \, \mathrm{d}u \leq \int_{\frac{u_0}{2}}^{u_0} g(u) \, \mathrm{d}u,$$

which implies that $\mathbb{P}\left(U \leq \frac{u_0}{2}\right) \leq \frac{1}{2}$. Then by Markov's inequality, we have

$$\mathbb{E}[U] \geq \frac{u_0}{2} \mathbb{P}\left(U > \frac{u_0}{2}\right) \geq \frac{u_0}{4},$$

which is just the desired result.

To this end, let $\mu = 1 > 0$ and $y = \frac{1}{u} + 1$, then

$$\frac{\mathrm{d}\ell}{\mathrm{d}u}(u) = \frac{1}{u} - 2Ku + \frac{2M}{u^2 y^3 \left(e^{\frac{1}{y^2}} - 1\right)} = \frac{1}{u} + 2Ku\left(\frac{M/K}{u^3 y^3 \left(e^{\frac{1}{y^2}} - 1\right)} - 1\right).$$

Since $1 \leq y$, when $u \leq \left(\frac{M}{K}\right)^{\frac{1}{3}} (e-1)^{-\frac{1}{3}} - 1$, we have

$$\frac{M/K}{u^3 y^3 \left(e^{\frac{1}{y^2}} - 1\right)} = \frac{M/K}{(u+1)^3 \left(e^{\frac{1}{y^2}} - 1\right)} \geq \frac{M/K}{(u+1)^3 (e-1)} \geq 1.$$

Take $C = (e-1)^{-\frac{1}{3}} - 2^{-\frac{1}{3}} > 0$ and then one can see that when $M \geq 2K$, $u_0 := C\left(\frac{M}{K}\right)^{1/3} \leq \left(\frac{M}{K}\right)^{\frac{1}{3}} (e-1)^{-\frac{1}{3}} - 1$. Then when $0 \leq u \leq u_0 := C\left(\frac{M}{K}\right)^{1/3}$, $\frac{\mathrm{d}\ell}{\mathrm{d}u}(u) \geq 0$. $\square$

# D Stability Term

In this section, we provide our results related to the stability term. We start with showing some important properties for $V_{i,N}$.

**Lemma D.1.** *For all $\mathcal{I} \subset \{1, \cdots, d\}$ such that $|\mathcal{I}| < m$ and $i \notin \mathcal{I}$, where $1 \leq i \leq d$, we have*

$$V_{i,N}^{\mathcal{I}}(\lambda) = \int_{-\min_{j \notin \mathcal{I}} \lambda_j}^{\infty} \frac{1}{(x+\lambda_i)^N} \prod_{q \in \mathcal{I}} (1 - F(x+\lambda_q)) \prod_{q \notin \mathcal{I}} F(x+\lambda_q) \, \mathrm{d}x.$$

*Then for all $1 \leq i \leq d$ such that $\sigma_i(\lambda) > m$, we have*

$$V_{i,N}(\lambda) = \int_{-\max_{\mathcal{I}, |\mathcal{I}| < m} \min_{j \notin \mathcal{I}} \lambda_j}^{\infty} \frac{1}{(x+\lambda_i)^N} e^{-1/(x+\lambda_i)^2} \sum_{s=0}^{m-1} \sum_{\mathcal{I} \subseteq \{1,\ldots,d\}\setminus\{i\}, |\mathcal{I}|=s} \left[\prod_{q \in \mathcal{I}} (1 - F(x+\lambda_q)) \prod_{q \notin \mathcal{I}, q \neq i} F(x+\lambda_q)\right] \mathrm{d}x.$$

*Proof.* For the first result, it suffices to note that $F(x) = 0$ when $x \leq 0$, then when $x \leq -\min_{j \notin \mathcal{I}} \lambda_j$, the integrand in Eq. (3) is just 0. For the second result, recall that

$$V_{i,N}(\lambda) = \sum_{s=0}^{m-1} \sum_{\mathcal{I} \subseteq \{1,\ldots,d\}\setminus\{i\}, |\mathcal{I}|=s} V_{i,N}^{\mathcal{I}}(\lambda)$$

and note that for all $\mathcal{I} \subset \{1, \cdots, d\}$ such that $|\mathcal{I}| < m$ and $i \notin \mathcal{I}$,

$$- \max_{\mathcal{I}, |\mathcal{I}| < m} \min_{j \notin \mathcal{I}} \lambda_j = \min_{\mathcal{I}, |\mathcal{I}| < m} \left( - \min_{j \notin \mathcal{I}} \lambda_j \right) \leq - \min_{j \notin \mathcal{I}} \lambda_j,$$

then the result follows from that for every $V_{i,N}^{\mathcal{I}}(\lambda)$, the lower limit of the integral can be further reduced to $- \max_{\mathcal{I}, |\mathcal{I}| < m} \min_{j \notin \mathcal{I}} \lambda_j$. $\qquad \square$

**Lemma D.2.** *The followings hold:*

1. *For all $N \geq 2$, $V_{i,N}(\lambda) \leq \frac{\underline{\lambda}_i^{1-N}}{N-1}$.*

2. *For all $1 \leq i \leq d$ and $N \geq 3$, $V_{i,N}(\lambda)$ is increasing in all $\lambda_j, j \neq i$ and decreasing in $\lambda_i$.*

3. *$\bar{V}_{i,N}(\lambda) := \frac{\Gamma(\frac{N-1}{2})}{2} - V_{i,N}(\lambda) =$*

$$\int_{-\lambda_i}^{\infty} \frac{1}{(x+\lambda_i)^N} e^{-1/(x+\lambda_i)^2} \sum_{s=m}^{d-1} \sum_{\mathcal{I} \subseteq \{1,\ldots,d\} \setminus \{i\}, |\mathcal{I}|=s} \left[ \prod_{q \in \mathcal{I}} (1 - F(x+\lambda_q)) \prod_{q \notin \mathcal{I}, q \neq i} F(x+\lambda_q) \right] \mathrm{d}x \geq 0.$$

*Proof.* For the first result, obviously, it suffices to consider the case when $\sigma_i(\lambda) > m$. By Lemma D.1, we have

$$V_{i,N}(\lambda) = \int_{- \max_{\mathcal{I}, |\mathcal{I}| < m} \min_{j \notin \mathcal{I}} \lambda_j}^{\infty} \frac{1}{(x+\lambda_i)^N} e^{-1/(x+\lambda_i)^2} \sum_{s=0}^{m-1} \sum_{\mathcal{I} \subseteq \{1,\ldots,d\} \setminus \{i\}, |\mathcal{I}|=s} \left[ \prod_{q \in \mathcal{I}} (1 - F(x+\lambda_q)) \prod_{q \notin \mathcal{I}, q \neq i} F(x+\lambda_q) \right] \mathrm{d}x$$

$$\leq \int_{- \max_{\mathcal{I}, |\mathcal{I}| < m} \min_{j \notin \mathcal{I}} \lambda_j}^{\infty} \frac{1}{(x+\lambda_i)^N} \mathrm{d}x = \int_0^{\infty} \frac{1}{(z+\underline{\lambda}_i)^N} \mathrm{d}z = \frac{\underline{\lambda}_i^{1-N}}{N-1},$$

where in the inequality, we upper bound the conditional probability inside (see Lemma C.3) by 1.

For the rest results, consider a random variable $r_i'$ with density

$$g(x) = \frac{x^{-N} e^{-1/x^2} \mathbb{1}_{\{x>0\}}}{\int_0^{+\infty} x^{-N} e^{-1/x^2} \mathrm{d}x} = \frac{2}{\Gamma(\frac{N-1}{2})} x^{-N} e^{-1/x^2} \mathbb{1}_{\{x>0\}}.$$

Then similar to Eq. (1),

$$V_{i,N}(\lambda) = \frac{\Gamma(\frac{N-1}{2})}{2} \mathbb{P}(\text{there exist at most } m-1 \text{ of } r_1 - \lambda_1, \cdots, r_{i-1} - \lambda_{i-1}, r_{i+1} - \lambda_{i+1}, \cdots,$$

$$r_d - \lambda_d \text{ that are larger than } r_i' - \lambda_i),$$

where $r_1, \cdots, r_{i-1}, r_{i+1}, \cdots, r_d \overset{\text{i.i.d.}}{\sim} \mathcal{F}_2$. Then these properties hold obviously. $\qquad \square$

**Lemma D.3.** *For all $1 \leq i \leq d$, we have*

$$\frac{V_{i,4}(\lambda)}{V_{i,3}(\lambda)} \leq \underline{\lambda}_i^{-1} \wedge C \begin{cases} \sqrt{\frac{m \log(d)}{\sigma_i(\lambda) - m}} & \sigma_i(\lambda) > 2m \\ m^{1/3} & \sigma_i(\lambda) \leq 2m, \end{cases}$$

*where $C$ is a positive constant.*

*Proof.* For any $\mathcal{I} \subseteq \{1, \cdots, d\}$ such that $|\mathcal{I}| < m$ and $i \notin \mathcal{I}$, by Lemma D.1, we have

$$V_{i,N}^{\mathcal{I}}(\lambda) = \int_{- \min_{j \notin \mathcal{I}} \lambda_j}^{\infty} \frac{1}{(x+\lambda_i)^N} \prod_{q \in \mathcal{I}} (1 - F(x+\lambda_q)) \prod_{q \notin \mathcal{I}} F(x+\lambda_q) \, \mathrm{d}x$$

$$= \int_0^{\infty} \frac{1}{(z + \underline{\lambda}_i^{\mathcal{I}})^N} \prod_{q \in \mathcal{I}} (1 - F(z + \underline{\lambda}_q^{\mathcal{I}})) \prod_{q \notin \mathcal{I}} F(z + \underline{\lambda}_q^{\mathcal{I}}) \, \mathrm{d}z, \tag{11}$$

where we denoted that $\underline{\lambda}_q^{\mathcal{I}} := \lambda_q - \min_{j \notin \mathcal{I}} \lambda_j$ for all $1 \leq q \leq d$ and we denoted that $z = x + \min_{j \notin \mathcal{I}} \lambda_j$.

Then, clearly, $\frac{V_{i,4}^{\mathcal{I}}(\lambda)}{V_{i,3}^{\mathcal{I}}(\lambda)} \leq (\underline{\lambda}_i^{\mathcal{I}})^{-1}$, which implies that

$$\frac{V_{i,4}(\lambda)}{V_{i,3}(\lambda)} \leq \max_{\mathcal{I},|\mathcal{I}|<m,i\notin\mathcal{I}} (\underline{\lambda}_i^{\mathcal{I}})^{-1} = (\lambda_i - \max_{\mathcal{I},|\mathcal{I}|<m,i\notin\mathcal{I}} \min_{j\notin\mathcal{I}} \lambda_j)^{-1} \leq \underline{\lambda}_i^{-1}.$$

For the second part, by Eq. (11) and Lemma 4.1, we have

$$\frac{V_{i,4}^{\mathcal{I}}(\lambda)}{V_{i,3}^{\mathcal{I}}(\lambda)} = \frac{J_{i,4,\mathcal{I}}(\underline{\lambda}^{\mathcal{I}})}{J_{i,N,\mathcal{I}}(\underline{\lambda}^{\mathcal{I}})} \leq \frac{J_{i,4,\mathcal{I}}(\lambda^\star)}{J_{i,N,\mathcal{I}}(\lambda^\star)} = \frac{\int_0^\infty \frac{1}{(z+\underline{\lambda}_i^{\mathcal{I}})^4} e^{-\frac{\sigma_i'(\lambda)}{(z+\underline{\lambda}_i^{\mathcal{I}})^2}} \prod_{q\in\mathcal{I}}(1 - F(z + \lambda_q^\star))\,\mathrm{d}z}{\int_0^\infty \frac{1}{(z+\underline{\lambda}_i^{\mathcal{I}})^3} e^{-\frac{\sigma_i'(\lambda)}{(z+\underline{\lambda}_i^{\mathcal{I}})^2}} \prod_{q\in\mathcal{I}}(1 - F(z + \lambda_q^\star))\,\mathrm{d}z}, \quad (12)$$

where we denoted that

$$\lambda_q^\star = \begin{cases} +\infty & q \notin \mathcal{I} \text{ and } \lambda_q \geq \lambda_i \\ \underline{\lambda}_i^{\mathcal{I}} & q \notin \mathcal{I} \text{ and } \lambda_q \leq \lambda_i \\ \underline{\lambda}_q^{\mathcal{I}} & q \in \mathcal{I}, \end{cases}$$

and $\lambda_i$ is the $\sigma_i'(\lambda)$-th smallest in $\{\lambda_q\}_{q\notin\mathcal{I}}$. To apply Lemma 4.2, one should note that here $K = \sigma_i'(\lambda)$ and $M = |\mathcal{I}|$. If $\sigma_i(\lambda) > 2m$, then $K \geq \sigma_i(\lambda) - m \geq m \geq M$, then by the second result in Lemma 4.2, there exists $C > 0$ such that the right hand in Eq. (12)

$$\leq C\sqrt{\frac{M}{K} \log\left(\frac{K}{M} + 1\right)} \leq C\sqrt{\frac{m\log(d)}{\sigma_i(\lambda) - m}}.$$

If $\sigma_i(\lambda) \leq 2m$, similarly, by the first result in Lemma 4.2, the right hand in Eq. (12) is less than

$$C\left(\frac{M}{K}\right)^{\frac{1}{3}} \leq m^{1/3}.$$

$\square$

**Lemma D.4.** *For all $1 \leq i \leq d$, $N \geq 3$ and $k \geq 1$, we have*

$$\frac{\bar{V}_{i,N+k}(\lambda)}{\bar{V}_{i,N}(\lambda)} \leq C_{N,k} d^{\frac{k}{3}},$$

*where $C_{N,k}$ is a positive constant.*

*Proof.* Recall that in Lemma D.2 we have

$$\int_{-\lambda_i}^\infty \frac{1}{(x+\lambda_i)^N} e^{-1/(x+\lambda_i)^2} \sum_{s=m}^{d-1} \sum_{\mathcal{I}\subseteq\{1,\ldots,d\}\setminus\{i\},|\mathcal{I}|=s} \left[\prod_{q\in\mathcal{I}}(1 - F(x + \lambda_q)) \prod_{q\notin\mathcal{I},q\neq i} F(x + \lambda_q)\right] \mathrm{d}x \geq 0,$$

then the proof follows from the case $\sigma_i(\lambda) \leq 2m$ in Lemma D.3 by noting that $|\mathcal{I}| \leq d$. $\square$

**Lemma D.5.** *There exists $C > 0$ such that for all $a \in [0, d^{-1}]$ and $1 \leq i \leq d$,*

$$\frac{\bar{V}_{i,6}(\lambda + ae_i)}{\bar{V}_{i,6}(\lambda)} \leq C.$$

*Proof.* It suffices to show that $\frac{\partial}{\partial\lambda_i} \log(\bar{V}_{i,6}(\lambda))$ is upper bounded by $Cd$. By the definition,

$$\frac{\partial}{\partial\lambda_i} \log(\bar{V}_{i,6}(\lambda)) = \frac{\frac{\partial}{\partial\lambda_i} \bar{V}_{i,6}(\lambda)}{\bar{V}_{i,6}(\lambda)},$$

where

$$\frac{\partial}{\partial\lambda_i} \bar{V}_{i,6}(\lambda) = \int_{-\lambda_i}^\infty \frac{\partial}{\partial\lambda_i}\left(\frac{1}{(x+\lambda_i)^6} e^{-1/(x+\lambda_i)^2}\right) \sum_{s=m}^{d-1} \sum_{\mathcal{I}\subseteq\{1,\ldots,d\}\setminus\{i\},|\mathcal{I}|=s} \left[\prod_{q\in\mathcal{I}}(1 - F(x + \lambda_q)) \prod_{q\notin\mathcal{I},q\neq i} F(x + \lambda_q)\right] \mathrm{d}x,$$

since $\lim_{x \to -\lambda_i} \frac{1}{(x+\lambda_i)^6} e^{-1/(x+\lambda_i)^2} = 0$ . Note that $\frac{\partial}{\partial \lambda_i} \left( \frac{1}{(x+\lambda_i)^6} e^{-1/(x+\lambda_i)^2} \right) \leq \frac{2}{(x+\lambda_i)^9} e^{-1/(x+\lambda_i)^2}$,
then it's clear that $\frac{\partial}{\partial \lambda_i} \bar{V}_{i,6}(\lambda) \leq 2\bar{V}_{i,9}(\lambda)$. Hence, by Lemma D.4, there exits $C > 0$ such that
$\frac{\partial}{\partial \lambda_i} \log(\bar{V}_{i,6}(\lambda)) \leq Cd$. $\qquad\qquad\qquad\qquad\qquad\qquad\qquad\qquad\qquad\qquad\qquad\qquad\qquad\qquad\square$

Then we show our the results about the continuity of $\phi$.

**Lemma D.6.** *There exists $C > 0$ such that for all $1 \leq i \leq d$, $a > 0$ and $\lambda \in \mathbb{R}^d$, if $w = \phi_i(\lambda)$ and $w' = \phi_i(\lambda + ae_i)$, then the followings hold:*

*1.* $w - w' \leq Cwa \cdot \underline{\lambda}_i^{-1} \wedge \begin{cases} \sqrt{\frac{m \log(d)}{\sigma_i(\lambda) - m}} & \sigma_i(\lambda) > 2m \\ m^{1/3} & \sigma_i(\lambda) \leq 2m. \end{cases}$

*2.* $w - w' \leq Cd(1 - w)a$, *if* $a \leq d^{-1}$.

*Proof.* For the first result, note that for all $t \in [0, 1]$
$$\frac{\mathrm{d}}{\mathrm{d}t} \phi_i(\lambda + (1 - t)ae_i) = -a \frac{\partial \phi_i}{\partial \lambda_i}(\lambda + (1 - t)ae_i).$$

Then,
$$-\frac{\partial \phi_i}{\partial \lambda_i}(\lambda + (1-t)ae_i) = 6V_{i,4}(\lambda + (1-t)ae_i) - 4V_{i,6}(\lambda + (1-t)ae_i) \leq 6V_{i,4}(\lambda + (1-t)ae_i) \leq 6V_{i,4}(\lambda),$$

where we used Lemma D.2 in the final inequality. Recall that $w = \phi_i(\lambda) = 2V_{i,3}(\lambda)$, then by Lemma D.3,
$$-\frac{\partial \phi_i}{\partial \lambda_i}(\lambda + (1 - t)ae_i) \leq C'w \cdot \underline{\lambda}_i^{-1} \wedge \begin{cases} \sqrt{\frac{m \log(d)}{\sigma_i(\lambda) - m}} & \sigma_i(\lambda) > 2m \\ m^{1/3} & \sigma_i(\lambda) \leq 2m. \end{cases}$$

Therefore,
$$w - w' = \int_0^1 \frac{\mathrm{d}}{\mathrm{d}t} \phi_i(\lambda + (1 - t)ae_i)\,\mathrm{d}t \leq C'wa \cdot \underline{\lambda}_i^{-1} \wedge \begin{cases} \sqrt{\frac{m \log(d)}{\sigma_i(\lambda) - m}} & \sigma_i(\lambda) > 2m \\ m^{1/3} & \sigma_i(\lambda) \leq 2m. \end{cases} \qquad (13)$$

For the second result, let $\bar{\phi} = \mathbf{1} - \phi$, then $w - w' = \bar{\phi}_i(\lambda + ae_i) - \bar{\phi}_i(\lambda)$. Similarly, for all $t \in [0, 1]$,
$$\frac{\mathrm{d}}{\mathrm{d}t} \bar{\phi}_i(\lambda + tae_i) = a \frac{\partial \bar{\phi}_i}{\partial \lambda_i}(\lambda + tae_i).$$

Since now $\bar{\phi}_i(\lambda) = 2\bar{V}_{i,3}(\lambda)$, then clearly,
$$\frac{\partial \bar{\phi}_i}{\partial \lambda_i}(\lambda) = -6\bar{V}_{i,4}(\lambda) + 4\bar{V}_{i,6}(\lambda) \leq 4\bar{V}_{i,6}(\lambda).$$

Hence, combing Lemma D.5 and Lemma D.4, we have
$$\frac{\partial \bar{\phi}_i}{\partial \lambda_i}(\lambda + tae_i) \leq 4\bar{V}_{i,6}(\lambda + tae_i) \leq C\bar{V}_{i,6}(\lambda) \leq C'd\bar{V}_{i,3}(\lambda) = C''d(1 - w).$$

Finally, one can obtain the result by the way similar to Eq. (13). $\qquad\qquad\qquad\qquad\square$

## D.1 Proof for Lemma 4.3

*Proof.* By Lemma D.2,
$$\phi_i(\eta_t \hat{L}_{t+1}) \geq \phi_i(\eta_t \hat{L}_t + \eta_t \hat{\ell}_{t,i} \cdot e_i),$$
then
$$\hat{\ell}_{t,i} \left( \phi_i(\eta_t \hat{L}_t) - \phi_i(\eta_t \hat{L}_{t+1}) \right) \leq \hat{\ell}_{t,i} \left( \phi_i(\eta_t \hat{L}_t) - \phi(\eta_t \hat{L}_t + \eta_t \hat{\ell}_{t,i} \cdot e_i) \right).$$
Denote that
$$\Lambda = \hat{\underline{L}}_{t,i}^{-1} \wedge \eta_t \begin{cases} \sqrt{\frac{m \log(d)}{\sigma_i(\hat{L}_t) - m}} & \sigma_i(\hat{L}_t) > 2m \\ m^{1/3} & \sigma_i(\hat{L}_t) \leq 2m. \end{cases}$$

Hence, by Lemma D.6

$$\hat{\ell}_{t,i}(w - w') \leq C\Lambda w \hat{\ell}_{t,i}^2 \leq C\Lambda w K_{t,i}^2 A_{t,i},$$

where we denoted that $w = \phi_i(\eta_t \hat{L}_t)$ and $w' = \phi_i(\eta_t \hat{L}_t + \eta_t \hat{\ell}_{t,i} \cdot e_i)$. By Lemma G.4, $\mathbb{E}[K_{t,i}^2 \mid \mathscr{F}_{t-1}, A_t] \leq 2w^{-2}$. Then

$$\mathbb{E}[\hat{\ell}_{t,i}(w - w') \mid \mathscr{F}_{t-1}, A_t] \leq 2C\Lambda \cdot w^{-1} A_{t,i}.$$

Hence, since $\mathbb{E}[A_{t,i} \mid \mathscr{F}_{t-1}] = w$, then

$$\mathbb{E}[\hat{\ell}_{t,i}(w - w') \mid \mathscr{F}_{t-1}] \leq 2C\Lambda.$$

$\square$

## D.2   Proof for Lemma 4.5

*Proof.* By Lemma 4.3, it's clear that

$$\sum_{i=m+1}^{d} \mathbb{E}\left[\hat{\ell}_{t,i}\left(\phi_i(\eta_t \hat{L}_t) - \phi_i(\eta_t \hat{L}_{t+1})\right) \mid \mathscr{F}_{t-1}\right] \leq C \sum_{i=m+1}^{d} \hat{L}_{t,i}^{-1},$$

then it suffices to tackle the sum for $1 \leq i \leq m$. By Lemma D.6 and following the same argument in Lemma 4.3, for all $1 \leq i \leq m$, we have

$$\mathbb{E}\left[\mathbb{1}_{\{\eta_t K_{t,i} \leq d^{-1}\}} \hat{\ell}_{t,i}\left(\phi_i(\eta_t \hat{L}_t) - \phi_i(\eta_t \hat{L}_{t+1})\right) \mid \mathscr{F}_{t-1}\right] \leq C\eta_t d w_{t,i}^{-1}(1 - w_{t,i}),$$

where we denoted that $w_t = \phi(\eta_t \hat{L}_t)$ and we used the fact that $\eta_t \hat{\ell}_{t,i} \leq d^{-1}$ when $\eta_t K_{t,i} \leq d^{-1}$. Note that by Lemma F.1, when $\sum_{i=m+1}^{d}(\eta_t \underline{\hat{L}}_{t,i})^{-2} < \frac{1}{2m}$, for all $1 \leq i \leq m$,

$$w_{t,i} \geq w_\star \geq 1/2. \tag{14}$$

Hence,

$$\sum_{i=1}^{m} \mathbb{E}\left[\mathbb{1}_{\{\eta_t K_{t,i} \leq d^{-1}\}} \hat{\ell}_{t,i}\left(\phi_i(\eta_t \hat{L}_t) - \phi_i(\eta_t \hat{L}_{t+1})\right) \mid \mathscr{F}_{t-1}\right] \leq 2C\eta_t d \sum_{i=1}^{m}(1 - w_{t,i}) = 2C\eta_t d \sum_{i=m+1}^{d} w_{t,i},$$

where we used that $\sum_{i=1}^{d} w_{t,i} = m$ by Lemma C.2. Finally, for all $1 \leq i \leq m$,

$$\mathbb{E}\left[\mathbb{1}_{\{\eta_t K_{t,i} > d^{-1}\}} \hat{\ell}_{t,i}\left(\phi_i(\eta_t \hat{L}_t) - \phi_i(\eta_t \hat{L}_{t+1})\right) \mid \mathscr{F}_{t-1}\right] \leq \mathbb{E}\left[\mathbb{1}_{\{\eta_t K_{t,i} > d^{-1}\}} A_{t,i} K_{t,i} \mid \mathscr{F}_{t-1}\right]$$

$$= \mathbb{E}\left[\mathbb{1}_{\{\eta_t K_{t,i} > d^{-1}\}} w_{t,i} K_{t,i} \mid \mathscr{F}_{t-1}\right],$$

which, by Lemma G.4, is less than

$$(1 - w_{t,i})^{\lfloor d^{-1} \eta_t^{-1} \rfloor} \leq (1 - w_{t,i})^{\frac{1}{2\eta_t d}} \leq 2^{-\frac{1}{2\eta_t d}},$$

where we used Eq. (14) in the final inequality. It suffices to combine everything together naively.   $\square$

# E   Penalty Term

In this section, we present our results related to the penalty term.

**Lemma E.1.** *If $r \sim \mathcal{F}_2$, then for all $x \geq 1$, we have*

$$\mathbb{E}[r \mid r \geq x] \leq 4x.$$

*Proof.* By the definition, we have

$$\mathbb{E}[r \mid r \geq x] = \frac{\int_x^{+\infty} u f(u) \, du}{1 - F(x)} \leq 2x^2 \int_x^{+\infty} u f(u) \, du \leq 2x^2 \int_x^{+\infty} 2u^{-2} \, du = 4x,$$

where the first inequality used that $1 - e^{-x} \geq x/2$ when $0 \leq x \leq 1$ and the second inequality used that $e^{-1/u^2} \leq 1$.   $\square$

**Lemma E.2.** *Consider* $r_1, \cdots, r_d \overset{\text{i.i.d.}}{\sim} \mathcal{F}_2$ *with* $r_{(k)}$ *as the kth order statistic for all* $1 \leq k \leq d$. *Then for all* $m \leq d$, *the expectation of the largest* $m$ *numbers, say* $\mathbb{E}\left[\sum_{k=d-m+1}^{d} r_{(k)}\right]$, *is less than* $5\sqrt{md}$.

*Proof.* Clearly,

$$\mathbb{E}\left[\sum_{k=d-m+1}^{d} r_{(k)}\right] \leq \sqrt{md} + \mathbb{E}\left[\sum_{k=d-m+1}^{d} r_{(k)} \cdot \mathbb{1}_{\left\{r_{(k)} \geq \sqrt{d/m}\right\}}\right] \leq \sqrt{md} + \sum_{k=1}^{d} \mathbb{E}\left[r_k \cdot \mathbb{1}_{\left\{r_k \geq \sqrt{d/m}\right\}}\right].$$

Then it suffices to note that, by Lemma E.1, we have

$$\mathbb{E}\left[r_k \cdot \mathbb{1}_{\left\{r_k \geq \sqrt{d/m}\right\}}\right] \leq 4\sqrt{d/m}\,\mathbb{P}\left(r_k \geq \sqrt{d/m}\right) = 4\sqrt{d/m}\left(1 - \mathrm{e}^{-m/d}\right) \leq 4\sqrt{m/d},$$

where the final inequality used that $1 - \mathrm{e}^{-x} \leq x$. $\qquad \square$

## E.1 Proof for Lemma 3.4

*Proof.* If $a = \nabla\Phi(\lambda)$, which implies that $a \in \partial\Phi(\lambda)$, then by Lemma G.1, we have

$$\Phi^*(a) = \langle\lambda, a\rangle - \Phi(\lambda) = \mathbb{E}[\langle\lambda, A\rangle] - \Phi(\lambda) = \mathbb{E}[\langle\lambda, A\rangle] - \mathbb{E}[\langle r + \lambda, A\rangle] = -\mathbb{E}[\langle r, A\rangle],$$

where we used that $\mathbb{E}[A] = \phi(-\lambda) = a$ in the second equality and the third equality follows from the definition of $\Phi$. If $a \in \mathcal{A}$, for all $x \in \mathbb{R}^d$, we have $\Phi(x) \geq \mathbb{E}[\langle r + x, a\rangle]$, which implies that

$$\Phi^*(a) \leq \sup_{x \in \mathbb{R}^d} \langle x, a\rangle - \mathbb{E}[\langle r + x, a\rangle] = -\mathbb{E}[\langle r, a\rangle].$$

$\qquad \square$

## E.2 Proof for Lemma 4.6

*Proof.* By Lemma 3.4,

$$\Phi^*(a) - \Phi^*(\phi(\lambda)) \leq \mathbb{E}[\langle r, A - a\rangle], \tag{15}$$

where $A = \arg\max_{a \in \mathcal{A}} \langle r + \lambda, a\rangle$. Then for the first result, since $r \in \mathbb{R}^{+d}$,

$$\Phi^*(a) - \Phi^*(\phi(\lambda)) \leq \mathbb{E}[\max_{a' \in \mathcal{A}} \langle r, a'\rangle],$$

which is less than $5\sqrt{md}$ by Lemma E.2.

For the second result, W.L.O.G., we assume that $\lambda_1 \leq \cdots \leq \lambda_d$, then $a = (\underbrace{1, \cdots, 1}_{m \text{ of } 1}, \underbrace{0, \cdots, 0}_{d-m \text{ of } 0})$ and by Eq. (15),

$$\Phi^*(a) - \Phi^*(\phi(\lambda)) \leq \sum_{i=m+1}^{d} \mathbb{E}[r_i \mathbb{1}_{\{A_i=1\}}].$$

By the definition of $A$, for all $i > m$, we have

$$\mathbb{E}[r_i \mathbb{1}_{\{A_i=1\}}] = \mathbb{E}_{r_i}[r_i \mathbb{E}[\mathbb{1}_{\{A_i=1\}} \mid r_i = x + \lambda_i]]$$

$$= \int_{-\lambda_i}^{\infty} \frac{2(x + \lambda_i)}{(x + \lambda_i)^3} e^{-1/(x+\lambda_i)^2} \sum_{s=0}^{m-1} \sum_{\mathcal{I} \subseteq \{1,\ldots,d\}\setminus\{i\}, |\mathcal{I}|=s} \left[\prod_{q \in \mathcal{I}} (1 - F(x + \lambda_q)) \prod_{q \notin \mathcal{I}, q \neq i} F(x + \lambda_q)\right] dx$$

$$= 2V_{i,2}(\lambda) \leq 2\underline{\lambda}_i^{-1},$$

where we used Lemma D.2 in the final inequality. This completes our proof. $\qquad \square$

# F    Probability of the Best Action

In this section, we present our results related to the lower bound for the regret.

**Lemma F.1.** *For all $\lambda \in \mathbb{R}^d$, let*

$$w_\star = \mathbb{P}\{\min_{1 \le i \le m} (r_i - \lambda_i) \ge \max_{m+1 \le i \le d} (r_i - \lambda_i)\},$$

*where $r_1, \cdots, r_d \overset{\text{i.i.d.}}{\sim} \mathcal{F}_2$. Then, we have the followings:*

1. *If $\sum_{i=m+1}^{d} \underline{\lambda}_i^{-2} < \frac{1}{2m}$, then $w_\star \ge \frac{1}{2}$.*

2. *If $\sum_{i=m+1}^{d} \underline{\lambda}_i^{-2} \ge \frac{1}{2m}$, then $w_\star \le 1 - \frac{1}{16m}$.*

*Proof.* If $\sum_{i=m+1}^{d} \underline{\lambda}_i^{-2} = +\infty$, then there exists $1 \le i \le m < j \le d$ such that $\lambda_i \ge \lambda_j$. W.L.O.G., we assume that $\lambda_m \ge \lambda_{m+1}$. Denote that $X_i = r_i - \lambda_i$ for all $1 \le i \le d$, then clearly,

$$w_\star = \mathbb{P}\{X_1, \cdots, X_m \text{ are the } m \text{ largest values among } X_1, \cdots, X_d\}$$
$$\le \mathbb{P}\{X_1, \cdots, X_{m-1}, X_{m+1} \text{ are the } m \text{ largest values among } X_1, \cdots, X_d\} := w'_\star.$$

Note that $w_\star + w'_\star \le 1$, then we have $w_\star \le 1/2 \le 1 - \frac{1}{16m}$.

Then we assume that $\sum_{i=m+1}^{d} \underline{\lambda}_i^{-2} < +\infty$, which implies that $\max_{1 \le i \le m} \lambda_i < \max_{m+1 \le i \le d} \lambda_i$. W.L.O.G., we assume that $\lambda_1 \le \cdots \le \lambda_m < \lambda_{m+1} \le \cdots \le \lambda_d$. Hence, $\max_{\mathcal{I}, |\mathcal{I}| < m} \min_{j \notin \mathcal{I}} \lambda_j = \lambda_m$ and $\underline{\lambda}_i = \lambda_i - \lambda_m$ for all $i > m$. By the definition of $w_\star$,

$$w_\star = 2 \int_{-\lambda_{m+1}}^{+\infty} \sum_{i=m+1}^{d} (x+\lambda_i)^{-3} e^{-\sum_{j=m+1}^{d}(x+\lambda_j)^{-2}} \prod_{1 \le k \le m} \left(1 - e^{-(x+\lambda_k)^{-2}} \mathbb{1}_{\{x+\lambda_k \ge 0\}}\right) dx. \quad (16)$$

We first prove the lower bound for $w_\star$. Since for all $k \le m$, $\lambda_k \le \lambda_m$, then

$$w_\star \ge 2 \int_{-\lambda_{m+1}}^{+\infty} \sum_{i=m+1}^{d} (x + \lambda_i)^{-3} e^{-\sum_{j=m+1}^{d}(x+\lambda_j)^{-2}} \left(1 - e^{-(x+\lambda_m)^{-2}} \mathbb{1}_{\{x+\lambda_m \ge 0\}}\right)^m dx.$$

Then by Bernoulli's inequality, we have

$$w_\star \ge 2 \int_{-\lambda_{m+1}}^{+\infty} \sum_{i=m+1}^{d} (x + \lambda_i)^{-3} e^{-\sum_{j=m+1}^{d}(x+\lambda_j)^{-2}} \left(1 - m e^{-(x+\lambda_m)^{-2}} \mathbb{1}_{\{x+\lambda_m \ge 0\}}\right) dx$$

$$= 2 \int_{-\lambda_{m+1}}^{+\infty} \sum_{i=m+1}^{d} (x + \lambda_i)^{-3} e^{-\sum_{j=m+1}^{d}(x+\lambda_j)^{-2}} dx - 2m \int_{-\lambda_m}^{+\infty} \sum_{i=m+1}^{d} (x + \lambda_i)^{-3} e^{-\sum_{j=m+1}^{d}(x+\lambda_j)^{-2}-(x+\lambda_m)^{-2}} dx$$

$$= 1 - m\left(1 - \int_{-\lambda_m}^{+\infty} 2(x + \lambda_m)^{-3} e^{-\sum_{j=m+1}^{d}(x+\lambda_j)^{-2}-(x+\lambda_m)^{-2}} dx\right)$$

$$= 1 - m\left(1 - \int_{0}^{+\infty} 2z^{-3} e^{-\sum_{j=m+1}^{d}(z+\underline{\lambda}_j)^{-2}-z^{-2}} dz\right)$$

$$\ge 1 - m\left(1 - e^{-\sum_{j=m+1}^{d} \underline{\lambda}_j^{-2}} \int_{0}^{+\infty} 2z^{-3} e^{-z^{-2}} dz\right)$$

$$= 1 - m\left(1 - e^{-\sum_{j=m+1}^{d} \underline{\lambda}_j^{-2}}\right),$$

where in the third line we used that $\mathrm{d}\left(\mathrm{e}^{-\sum\limits_{j=m+1}^{d}(x+\lambda_j)^{-2}}\right)=2\sum\limits_{i=m+1}^{d}(x+\lambda_i)^{-3}\mathrm{e}^{-\sum\limits_{j=m+1}^{d}(x+\lambda_j)^{-2}}$

and in the fourth line we used that $\underline{\lambda}_i=\lambda_i-\lambda_m$ for all $i>m$. Hence, when $\sum\limits_{i=m+1}^{d}\underline{\lambda}_i^{-2}<\frac{1}{2m}$, by that $-1+\mathrm{e}^{-x}\geq x$, we have

$$w_\star\geq1-\frac{m}{2m}=\frac{1}{2}.$$

We then show the upper bounds. By Eq. (16), clearly,

$$
\begin{aligned}
w_\star\leq&2\int_{-\lambda_{m+1}}^{+\infty}\sum_{i=m+1}^{d}(x+\lambda_i)^{-3}\mathrm{e}^{-\sum\limits_{j=m+1}^{d}(x+\lambda_j)^{-2}}\left(1-\mathrm{e}^{-(x+\lambda_m)^{-2}}\mathbb{1}_{\{x+\lambda_m\geq0\}}\right)\mathrm{d}x\\
=&1-2\int_{-\lambda_m}^{+\infty}\sum_{i=m+1}^{d}(x+\lambda_i)^{-3}\mathrm{e}^{-\sum\limits_{j=m+1}^{d}(x+\lambda_j)^{-2}-(x+\lambda_m)^{-2}}\mathrm{d}x\\
=&1-\int_{0}^{+\infty}2\sum_{i=m+1}^{d}(z+\underline{\lambda}_i)^{-3}\mathrm{e}^{-\sum\limits_{j=m+1}^{d}(z+\underline{\lambda}_j)^{-2}-z^{-2}}\mathrm{d}z\\
\leq&1-\int_{\underline{\lambda}_{m+1}}^{+\infty}2\sum_{i=m+1}^{d}(z+\underline{\lambda}_i)^{-3}\mathrm{e}^{-\sum\limits_{j=m+1}^{d}(z+\underline{\lambda}_j)^{-2}-z^{-2}}\mathrm{d}z.
\end{aligned}
\tag{17}
$$

If $\sum\limits_{i=m+1}^{d}\underline{\lambda}_i^{-2}\geq\frac{1}{2m}$, note that when $z\geq\underline{\lambda}_{m+1}$, $\frac{1}{z^2}\leq\frac{4}{(z+\underline{\lambda}_{m+1})^2}$, then

$$\sum_{j=m+1}^{d}(z+\underline{\lambda}_j)^{-2}+z^{-2}\leq5\sum_{j=m+1}^{d}(z+\underline{\lambda}_j)^{-2}.$$

Hence, by Eq. (17),

$$
\begin{aligned}
w_\star\leq&1-\int_{\underline{\lambda}_{m+1}}^{+\infty}2\sum_{i=m+1}^{d}(z+\underline{\lambda}_i)^{-3}\mathrm{e}^{-5\sum\limits_{j=m+1}^{d}(z+\underline{\lambda}_j)^{-2}}\mathrm{d}z\\
=&1-\frac{1}{5}\left(1-\mathrm{e}^{-5\sum\limits_{j=m+1}^{d}(\underline{\lambda}_{m+1}+\underline{\lambda}_j)^{-2}}\right)\\
\leq&1-\frac{1}{5}\left(1-\mathrm{e}^{-\frac{5}{4}\sum\limits_{j=m+1}^{d}\underline{\lambda}_j^{-2}}\right)\\
\leq&1-\frac{1}{5}\left(1-\mathrm{e}^{-\frac{5}{8m}}\right)\leq1-\frac{1}{16m},
\end{aligned}
$$

where we used that $1-\mathrm{e}^{-x}\geq\frac{x}{2}$ when $x\in[0,1]$ in the final inequality. $\qquad\square$

**Lemma F.2.** *If $\sum\limits_{i=m+1}^{d}\underline{\lambda}_i^{-2}<\frac{1}{2m}$, then for all $m<i\leq d$, $\phi_i(\lambda)\geq\frac{1}{4\mathrm{e}}\underline{\lambda}_i^{-2}$.*

*Proof.* By the definition of $\phi_i$ in Eq. (1), clearly, for all $m<i\leq d$,

$$
\begin{aligned}
\phi_i(\lambda)\geq&\mathbb{P}\{\,r_i-\lambda_i\text{ is the largest in }r_m-\lambda_m,\cdots,r_d-\lambda_d\,\}\\
=&\int_{-\min\limits_{m\leq j\leq d}\lambda_j}^{\infty}\frac{2}{(z+\lambda_i)^3}\exp\left(-\sum_{j=m}^{d}\frac{1}{(z+\lambda_j)^2}\right)\mathrm{d}z.
\end{aligned}
\tag{18}
$$

Since $\sum_{i=m}^{d} \underline{\lambda}_i^{-2} < \frac{1}{2m}$, similar to Lemma F.1, W.L.O.G., we assume that $\lambda_1 \leq \cdots \leq \lambda_m < \lambda_{m+1} \leq \cdots \leq \lambda_d$. Hence, $\max_{\mathcal{I},|\mathcal{I}|<m} \min_{j \notin \mathcal{I}} \lambda_j = \lambda_m$ and $\underline{\lambda}_i = \lambda_i - \lambda_m$ for all $i > m$. Therefore, by Eq. (18), we have

$$\phi_i(\lambda) \geq \int_0^\infty \frac{2}{(z + \underline{\lambda}_i)^3} \exp\left(-\sum_{j=m}^d \frac{1}{(z + \underline{\lambda}_j)^2}\right) \mathrm{d}z$$

$$\geq \int_{\underline{\lambda}_{m+1}}^\infty \frac{2}{(z + \underline{\lambda}_i)^3} \exp\left(-\sum_{j=m}^d \frac{1}{(z + \underline{\lambda}_j)^2}\right) \mathrm{d}z.$$

Note that when $z \geq \underline{\lambda}_{m+1}$,

$$\sum_{j=m}^d \frac{1}{(z + \underline{\lambda}_j)^2} = \sum_{j=m+1}^d (z + \underline{\lambda}_j)^{-2} + z^{-2} \leq \sum_{j=m+1}^d \underline{\lambda}_j^{-2} + \underline{\lambda}_{m+1}^{-2} \leq 2 \sum_{j=m+1}^d \underline{\lambda}_j^{-2} < \frac{1}{m} \leq 1,$$

then by Eq. (17),

$$\phi_i(\lambda) \geq \mathrm{e}^{-1} \int_{\underline{\lambda}_{m+1}}^{+\infty} 2(z + \underline{\lambda}_i)^{-3} \, \mathrm{d}z = \mathrm{e}^{-1} \left(\underline{\lambda}_i + \underline{\lambda}_{m+1}\right)^{-2} \geq \frac{1}{4\mathrm{e}} \underline{\lambda}_i^{-2},$$

where in the final inequality we used that $\underline{\lambda}_{m+1} \leq \underline{\lambda}_i$ for all $i \geq m + 1$. $\qquad \square$

**Lemma F.3.** *Use the definition of $w_\star$ in Lemma F.1, then we have*

$$1 - w_\star \leq \sum_{i=m+1}^d \phi_i(\lambda).$$

*Proof.* Clearly,

$$1 - w_\star = \mathbb{P}\{\min_{1 \leq i \leq m} (r_i - \lambda_i) < \max_{m+1 \leq i \leq d} (r_i - \lambda_i)\}$$

$$= \mathbb{P}\left\{\bigcup_{i=m+1}^d \{r_i - \lambda_i \text{ is among the top } m \text{ largest values in } r_1 - \lambda_1, \cdots, r_d - \lambda_d\}\right\}$$

$$\leq \sum_{i=m+1}^d \phi_i(\lambda).$$

$\qquad \square$

# G  Auxiliary Lemma

**Lemma G.1** (Theorem 23.5 in Pryce [1973]). *For any continuous convex function $g$ and any vector $x$, the following conditions on a vector $x^*$ are equivalent to each other:*

*1. $x^* \in \partial g(x)$.*

*2. $x \in \partial g^*(x^*)$.*

*3. $g(x) + g^*(x^*) = \langle x, x^* \rangle$.*

**Lemma G.2** (Generalized Pythagoras Identity).

$$D_\Phi(x, y) + D_\Phi(z, x) - D_\Phi(z, y) = \langle \nabla \Phi(x) - \nabla \Phi(y), x - z \rangle.$$

*Proof.* It suffices to expand the left hand by the definition that

$$D_\Phi(x, y) = \Phi(x) - \Phi(y) - \langle x - y, \nabla \Phi(y) \rangle.$$

$\qquad \square$

**Lemma G.3.** *If $u = \nabla\Phi(x)$ and $v = \nabla\Phi(y)$, then*

$$D_\Phi(y, x) = \Phi^*(u) - \Phi^*(v) - \langle u - v, y \rangle.$$

**Remark G.1.** *Informally, this is just the folklore that $D_\Phi(y, x) = D_{\Phi^*}(u, v)$. However, it remains unproven whether $\Phi^*$ is differentiable everywhere, making it inconvenient to discuss its Bregman divergence directly.*

*Proof.* By Lemma G.1, we have

$$\Phi^*(u) = \langle u, x \rangle - \Phi(x), \Phi^*(v) = \langle v, y \rangle - \Phi(y).$$

Then the right hand equals

$$\Phi(y) - \Phi(x) - \langle u, y - x \rangle = \Phi(y) - \Phi(x) - \langle \nabla\Phi(x), y - x \rangle = D_\Phi(y, x).$$

$\square$

**Lemma G.4.** *If $K$ is sampled from the Geometric distribution with parameter $p \in (0, 1)$, then $\mathbb{E}[K^2] \leq \frac{2}{p^2}$. Furthermore, for all $n \in \mathcal{N}^+$, $\mathbb{E}[K - K \wedge n] = p^{-1}(1 - p)^n$.*

*Proof.* For the first result,

$$\mathbb{E}[K^2] = \mathbb{E}[K]^2 + \mathrm{Var}(K) = \frac{1}{p^2} + \frac{1 - p}{p^2} \leq \frac{2}{p^2}.$$

For the second result, by direct calculation, we have

$$\mathbb{E}[K - K \wedge n] = \sum_{k=n+1}^{+\infty} \mathbb{P}(K \geq k) = \sum_{k=n+1}^{+\infty} (1 - p)^{k-1} = p^{-1}(1 - p)^n.$$

$\square$

**Lemma G.5.** *For all $n \in \mathcal{N}^+$,*

$$\sum_{k=1}^{n} \frac{1}{\sqrt{k}} \leq 2\sqrt{n}.$$

*Proof.*

$$\sum_{k=1}^{n} \frac{1}{\sqrt{k}} \leq 1 + \int_1^n \frac{1}{\sqrt{x}} \, dx \leq 2\sqrt{n}.$$

$\square$

**Lemma G.6.** *For all $x \in (0, 1]$, let $f(x) = \sum_{t=1}^{+\infty} 2^{-\sqrt{t}x}$, then there exists $C > 0$ such that $f(x) \leq \frac{C}{x^2}$.*

*Proof.* It suffices to note that

$$f(x) \leq \int_0^{+\infty} 2^{-\sqrt{t}x} \, dt = 2 \int_0^{+\infty} u 2^{-ux} \, du = \frac{2}{x^2} \int_0^{+\infty} v 2^{-v} \, dv.$$

$\square$

**Lemma G.7.** *For all $\mu \in \mathbb{R}$ and $y \geq x > 0$, if $y \geq 1$, then we have*

$$\frac{1 - F(x + \mu)}{1 - F(y + \mu)} \leq 8y^2.$$

*Proof.* If $x + \mu \leq y$, then

$$\frac{1 - F(x + \mu)}{1 - F(y + \mu)} \leq \frac{1}{1 - F(y + \mu)}.$$

When $y + \mu \leq 0$, the right hand equals 1 and is clearly less than $8y^2$. Otherwise, $y + \mu > 0$ and then it suffices to note that

$$1 - F(y + \mu) = 1 - e^{-(y+\mu)^{-2}} \geq 1 - e^{-y^{-2}/4} \geq y^{-2}/8,$$

where we used $y + \mu \leq y + x + \mu \leq 2y$ in the first inequality and $1 - e^{-x} \geq x/2$ when $0 \leq x \leq 1$ in the second inequality.

If $x + \mu > y$, then $y + \mu \geq x + \mu > y \geq 1$ and similarly,

$$1 - F(y + \mu) = 1 - e^{-(y+\mu)^{-2}} \geq (y + \mu)^{-2}/2.$$

Also, since for all $x \geq 0$, $1 - e^{-x} \leq x$, we then have

$$1 - F(x + \mu) = 1 - e^{-(x+\mu)^{-2}} \leq (x + \mu)^{-2}.$$

Therefore,

$$\frac{1 - F(x + \mu)}{1 - F(y + \mu)} \leq 2 \left( \frac{y + \mu}{x + \mu} \right)^2 < 2 \left( \frac{y + y - x}{y} \right)^2 \leq 8 \leq 8y^2.$$

$\square$

**Lemma G.8** (Theorem 1.2.6 in Durrett [2019])**.** *For $x > 0$,*

$$\left( x^{-1} - x^{-3} \right) \exp(-x^2/2) \leq \int_x^\infty \exp(-y^2/2) \, \mathrm{d}y \leq x^{-1} \exp(-x^2/2).$$

**Lemma G.9.** *If $Y \sim \mathcal{N}(0, \sigma^2)$ and $\mu \geq \sigma > 0$, then for all $k \in \mathcal{N}$, we have*

$$\mathbb{E}[Y^k \mathbb{1}_{\{Y \geq \mu\}}] \leq k!! \cdot \sigma \mu^{k-1} e^{-\frac{\mu^2}{2\sigma^2}} / \sqrt{2\pi},$$

*where we define that $0!! = 1$.*

*Proof.* If $k = 0$, by Lemma G.8, we have

$$\mathbb{E}[\mathbb{1}_{\{Y \geq \mu\}}] = \mathbb{E}[\mathbb{1}_{\{\frac{Y}{\sigma} \geq \frac{\mu}{\sigma}\}}] \leq \frac{\sigma}{\mu} e^{-\frac{\mu^2}{2\sigma^2}} / \sqrt{2\pi}.$$

If $k = 1$,

$$\mathbb{E}[Y \mathbb{1}_{\{Y \geq \mu\}}] = \frac{1}{\sqrt{2\pi\sigma^2}} \int_\mu^{+\infty} y e^{-\frac{y^2}{2\sigma^2}} \, \mathrm{d}y = \frac{1}{\sqrt{2\pi\sigma^2}} \sigma^2 e^{-\frac{\mu^2}{2\sigma^2}}.$$

Assume that the statement holds for all integers $0, 1, \ldots, k - 1$, then

$$\mathbb{E}[Y^k \mathbb{1}_{\{Y \geq \mu\}}] = \frac{1}{\sqrt{2\pi\sigma^2}} \int_\mu^{+\infty} y^k e^{-\frac{y^2}{2\sigma^2}} \, \mathrm{d}y = \frac{\sigma^2}{\sqrt{2\pi\sigma^2}} \int_\mu^{+\infty} y^{k-1} \, \mathrm{d}\left( -e^{-\frac{y^2}{2\sigma^2}} \right)$$

$$= \frac{\sigma^2}{\sqrt{2\pi\sigma^2}} \mu^{k-1} e^{-\frac{\mu^2}{2\sigma^2}} + (k-1)\sigma^2 \mathbb{E}[Y^{k-2} \mathbb{1}_{\{Y \geq \mu\}}].$$

Then it suffices to note that $\sigma \leq \mu$ and $1 + (k-1) \cdot (k-2)!! \leq k \cdot (k-2)!! = k!!$. $\square$

**Lemma G.10.** *For any random variable $X$, let its density be denoted by $f(x)$. If there exist $\mu \geq 2\sigma > 0$ and $\mu' > \mu$ such that*

$$\frac{\mathrm{d}\log f}{\mathrm{d}x}(x) \leq -x/\sigma^2, \quad \forall \mu' \geq x \geq \mu,$$

*then for all $k \in \mathcal{N}^+$, if $\mu' \geq 2\mu$, we have*

$$\mathbb{E}[X^k \mathbb{1}_{\{\mu' \geq X \geq \mu\}}] \leq 2k!! \cdot \mu^k.$$

*Proof.* Consider $Y \sim \mathcal{N}(0, \sigma^2)$, then we will show that

$$\mathbb{E}[X^k \, \mathbb{1}_{\{\mu' \geq X \geq \mu\}}] \leq \mathbb{E}[Y^k \,|\, \mu' \geq Y \geq \mu], \tag{19}$$

which just equals

$$\frac{\mathbb{E}[Y^k \mathbb{1}_{\{\mu' \geq Y \geq \mu\}}]}{\mathbb{P}(\mu' \geq Y \geq \mu)} \leq \frac{\mathbb{E}[Y^k \mathbb{1}_{\{Y \geq \mu\}}]}{\mathbb{P}(\mu' \geq Y \geq \mu)} = \frac{\sqrt{2\pi}\mathbb{E}[Y^k \mathbb{1}_{\{Y \geq \mu\}}]}{\int_{\frac{\mu}{\sigma}}^{\frac{\mu'}{\sigma}} e^{-\frac{x^2}{2}} \, \mathrm{d}x} \leq \frac{k!! \cdot \sigma \mu^{k-1} \mathrm{e}^{-\frac{\mu^2}{2\sigma^2}}}{\int_{\frac{\mu}{\sigma}}^{\frac{\mu'}{\sigma}} e^{-\frac{x^2}{2}} \, \mathrm{d}x}, \tag{20}$$

where we applied Lemma G.9 in the last inequality. By Lemma G.8, we have

$$\int_{\frac{\mu}{\sigma}}^{\frac{\mu'}{\sigma}} e^{-\frac{x^2}{2}} \, \mathrm{d}x \geq \left(\frac{\sigma}{\mu} - \frac{\sigma^3}{\mu^3}\right) e^{-\frac{\mu^2}{2\sigma^2}} - \frac{\sigma}{\mu'} e^{-\frac{\mu'^2}{2\sigma^2}} \geq \left(\frac{\sigma}{\mu} - \frac{\sigma^3}{\mu^3}\right) e^{-\frac{\mu^2}{2\sigma^2}} - \frac{\sigma}{2\mu} e^{-\frac{2\mu^2}{\sigma^2}}, \tag{21}$$

where we used $\mu' \geq 2\mu$ in the last inequality. Since $\mu \geq \sigma$, we have $e^{-\frac{2\mu^2}{\sigma^2}} \leq \frac{1}{2} e^{-\frac{\mu^2}{2\sigma^2}}$. Hence, combining Eq. (20) and Eq. (21), we have

$$\mathbb{E}[Y^k \,|\, \mu' \geq Y \geq \mu] \leq \frac{k!! \cdot \sigma \mu^{k-1} \mathrm{e}^{-\frac{\mu^2}{2\sigma^2}}}{\left(\frac{3\sigma}{4\mu} - \frac{\sigma^3}{\mu^3}\right) e^{-\frac{\mu^2}{2\sigma^2}}} \leq \frac{2k!! \cdot \mu^k}{\frac{3}{2} - \frac{2\sigma^2}{\mu^2}}.$$

The right hand is clearly less than $2\mu^k$ when $\mu \geq 2\sigma$. To show Eq. (19), it suffices to show that for all $0 \leq t \leq \mu'$,

$$\mathbb{P}(X \, \mathbb{1}_{\{\mu' \geq X \geq \mu\}} \geq t) \leq \mathbb{P}(Y \geq t \,|\, \mu' \geq Y \geq \mu), \tag{22}$$

which holds when $t \leq \mu$ because the right hand becomes 1 by the definition. In fact, we can prove a stronger result: for all $\mu' \geq t > \mu$,

$$\frac{\mathbb{P}(X \, \mathbb{1}_{\{\mu' \geq X \geq \mu\}} \geq t)}{\mathbb{P}(X \, \mathbb{1}_{\{\mu' \geq X \geq \mu\}} \geq \mu)} \leq \frac{\mathbb{P}(Y \geq t \,|\, \mu' \geq Y \geq \mu)}{\mathbb{P}(Y \geq \mu \,|\, \mu' \geq Y \geq \mu)}. \tag{23}$$

If there exists $t_0 > \mu$ that violates Eq. (23), then we have $\frac{\mathbb{P}(X \, \mathbb{1}_{\{\mu' \geq X \geq \mu\}} \geq t_0)}{\mathbb{P}(X \, \mathbb{1}_{\{\mu' \geq X \geq \mu\}} \geq \mu)} > \frac{\mathbb{P}(Y \geq t_0 \,|\, \mu' \geq Y \geq \mu)}{\mathbb{P}(Y \geq \mu \,|\, \mu' \geq Y \geq \mu)}$, which is equivalent to

$$\frac{\mathbb{P}(X \, \mathbb{1}_{\{\mu' \geq X \geq \mu\}} \geq t_0)}{\mathbb{P}(\mu \leq X \, \mathbb{1}_{\{\mu' \geq X \geq \mu\}} < t_0)} > \frac{\mathbb{P}(Y \geq t_0 \,|\, \mu' \geq Y \geq \mu)}{\mathbb{P}(Y < t_0 \,|\, \mu' \geq Y \geq \mu)},$$

because $\frac{x}{x+1}$ increases with positive $x$. Then we have

$$\frac{\int_{t_0}^{\mu'} f(x) \, \mathrm{d}x}{\int_{\mu}^{t_0} f(x) \, \mathrm{d}x} > \frac{\int_{t_0}^{\mu'} e^{-\frac{x^2}{2\sigma^2}} \, \mathrm{d}x}{\int_{\mu}^{t_0} e^{-\frac{x^2}{2\sigma^2}} \, \mathrm{d}x}. \tag{24}$$

By homogeneity, we may assume $f(t_0) = e^{-\frac{t_0^2}{2\sigma^2}}$. Now since for all $\mu' \geq x \geq \mu$, we have $\frac{\mathrm{d} \log f(x)}{\mathrm{d}x} \leq -x/\sigma^2$, then for all $x \geq t_0$, $f(x) \leq e^{-\frac{x^2}{2\sigma^2}}$; for all $\mu \leq x < t_0$, $f(x) \geq e^{-\frac{x^2}{2\sigma^2}}$, which contradicts with Eq. (24). $\qquad \square$

