# OpenReview forum: "Follow-the-Perturbed-Leader Nearly Achieves Best-of-Both-Worlds for the m-Set Semi-Bandit Problems"
_NeurIPS.cc/2025/Conference — NeurIPS 2025 poster_

### Official Review · Reviewer_1er3 · 2025-06-24

**Clarity:** 2
**Significance:** 4
**Originality:** 3
**Rating:** 4
**Confidence:** 3

**Summary:**

This paper studies the best-of-both-worlds (BOBW) achievability of the Follow-the-Perturbed-Leader (FTPL) algorithm in the combinatorial semi-bandit setting. The authors demonstrate that FTPL achieves near-optimal regret in adversarial environments and logarithmic regret in stochastic settings. While similar results have been established for FTRL-based approaches, this is the first result of its kind for FTPL. This contribution is particularly valuable, as FTPL does not require solving complex optimization problems, which can be computationally expensive in combinatorial settings.

**Questions:**

Regarding the weaknesses, I would like to point out a recent note by Chen and Honda (2025), which argues that the current proof of Lemma 4.1 is incomplete and contains a technical flaw. Furthermore, their numerical validation suggests that the current argument may not hold as intended. I am curious whether the authors believe the proof can be modified or extended to rigorously establish the desired $\sqrt{nmd\log d}$ regret bound.

I would also like to emphasize that I fully acknowledge both works were developed independently, and the positive results presented by Chen and Honda have no influence on my overall evaluation. My assessment is based entirely on the concerns about the proof of Lemma 4.1, which plays a critical role in bounding the stability term and supporting the main theoretical results.

---
Chen, Botao, and Junya Honda. "Note on Follow-the-Perturbed-Leader in Combinatorial Semi-Bandit Problems." arXiv preprint arXiv:2506.12490 (2025).

**Ethical Concerns:**

["NO or VERY MINOR ethics concerns only"]

**Final Justification:**

During the rebuttal, the authors provided a revised analysis to address the issue in Lemma 4.1.
At least for the new Lemma 4.2, the analysis appears correct, and the updated regret becomes
$$ O(\sqrt{nmd\log d} + m^{4/3}\sqrt{n}). $$
Therefore, when $(d \log d)^{\frac{3}{5}} \leq m \leq d/2$, the adversarial regret becomes worse than expected. However, as the authors noted, it cannot be improved within the current proof framework.

Nevertheless, this work presents the first BOBW guarantee for FTPL under the combinatorial semi-bandit setting, where the adversarial regret is at least an improvement over the previous FTPL bound of $O(m\sqrt{nd\log (d/m)})$, while also achieving logarithmic stochastic regret. Although I expect that the adversarial regret can be improved further, doing so would likely require a different type of analysis or even a different choice of perturbations, which would be beyond the scope of this paper.

Although the current rebuttal policy makes it difficult to fully verify every detail, I think the modified proof is convincing. Therefore, I will raise my score to 4. Although I cannot change my confidence score, while lowering my confidence to 3.

**Limitations:**

yes

**Quality:**

1

**Strengths And Weaknesses:**

### Strength

This paper presents the first BOBW result for FTPL in combinatorial settings, along with improved adversarial regret bounds. Specifically, the authors achieve a regret of $\sqrt{nmd\log d}$ improving upon the previous bound of $m\sqrt{nd\log(d/m)}$ established by Neu and Bartók (2016).
Although several FTRL-based approaches have achieved optimal adversarial regret (without logarithmic dependencies), they typically require solving complex optimization problems to compute arm-selection probabilities at each round. In combinatorial settings, these optimization steps become significantly more computationally demanding compared to standard multi-armed bandit problems. In this regard, achieving BOBW guarantees with FTPL, an algorithm that avoids solving such optimization problems, is particularly appealing and makes the contribution of this paper especially valuable.

---

### Weakness

In the current analysis of the stability term, Lemma 4.1 plays a key role, establishing the foundation for Lemmas 4.2, 4.3, and 4.4, which collectively address the main analytical challenges in studying FTPL. While the paper provides a detailed proof showing that $J_{i,N+k,I}/J_{i,N,I}$ is increasing with respect to $\lambda_q \geq 0$ for $q\not\in I$, the corresponding decreasing result for $q\in I$ is not provided. The authors simply state that similar arguments would suffice, but I believe this step is not straightforward and may not follow from analogous reasoning.

This is because, as noted by the authors in Line 193, in the multi-armed bandit setting, the corresponding terms do not involve $1-F$ factors.
This structural difference becomes crucial when considering the derivative with respect to $\lambda_q$ for $q\in I$, where the presence of $1-F$ terms introduces additional dependencies that were not present in the multi-armed case and $q\not\in I$ case.
THerefore, the decreasing property for $q\in I$ may require a more careful, explicit argument rather than a direct extension of the existing proof.

To be more precise, fix $s \in I$.
Then, the partial derivatives of $J_{i,N,I}$ with respect to $\lambda_s$ is
$$
J_{i,N,I}^{s}(\lambda) := -\frac{1}{2}\frac{\partial J_{i,N,I}(\lambda)}{\partial \lambda_s} =\int_0^\infty \frac{e^{-1/(z+\lambda_s)^2}}{(z+\lambda_i)^N(z+\lambda_s)^3} \prod_{q \in I \setminus \\{s\\}}(1-F(z+\lambda_q)) \prod_{q \not\int I} F(z+\lambda_q)  dz.
$$
Then, similarly to Line 499,
$$
\frac{\partial}{\partial \lambda_s} \frac{J_{i,N+k,I}(\lambda)}{J_{i,N,I}(\lambda)} = -2 \cdot \frac{J_{i,N+k,I}^s(\lambda)J_{i,N,I}(\lambda)-J_{i,N+k,I}(\lambda)J_{i,N,I}^s(\lambda)}{J_{i,N,I}(\lambda)^2}.
$$
While the exponential term can be absorbed into the $Q$ term (as defined in Line 496) when $s\not\in I$, since $F$ is expressed in terms of those exponential terms, the same absorption is not possible under the current definition of $Q$ when $s\in I$.
In this case, the exponential term must be explicitly included in the product involving $1-F$.
Therefore, it is necessary to carefully reconsider the analysis for the $s\in I$ case by defining
$$
Q(z)=\frac{1}{(z+\lambda_i)^N}\prod_{q \in I \setminus \\{s\\}}(1-F(z+\lambda_q)) \prod_{q \not\in I} F(z+\lambda_q).
$$
Then, if we follow the same steps, we need to consider the sign of
$$
\frac{e^{-1/(z+\lambda_s)^2}}{(z+\lambda_s)^3(1-e^{-1/(z+\lambda_s)^2})}-\frac{e^{-1/(w+\lambda_s)^2}}{(w+\lambda_s)^3(1-e^{-1/(w+\lambda_s)^2})}.
$$
Deriving such results is not straightforward, and therefore the authors should provide detailed proofs to support these claims.

---

> ### Author Rebuttal · Authors · 2025-07-30
>
> Thank you for your valuable review and constructive suggestions. Regarding the weaknesses and questions, we provide the following detailed responses:
>
> **Weakness**:
> We must acknowledge our oversight: the second half of **Lemma 4.1** was indeed incorrect. To address this issue, we have **strengthened Lemma 4.2** by **removing the assumption $\mu_i \leq \mu$**, which eliminates the need for the monotonicity argument used in the original Lemma 4.1.
>
> >Lemma 4.2 (**modified**):  For all $\mu\ge0$, $K\ge 1$, $ N\ge 3$ and $M\ge 1$, let $\mu_i\in R$ for all $1\leq i\leq M$, and define
>     $$
>     H_{N}=\int_0^{+\infty}(x+\mu)^{-N}e^{-\frac{K}{(x+\mu)^2}}\prod_{i=1}^M\left(1-F(x+\mu_i)\right) d x.
>     $$
>     For all $k\in\mathcal{N}^+$, we have
>     $$
>     \frac{H_{N+k}}{H_N}\leq C_{N,k}\left(\left(\frac{M}{K}\right)^{k/3}\wedge \mu^{-k}\right),
>     $$
>     where $C_{N,k}$ is a positive constant only depending on $N$ and $k$. Furthermore, if $K\ge M$, then we have
>     $$
>     \frac{H_{4}}{H_3}\leq C\left(\left(\frac{M}{K}\log\left(\frac{K}{M}+1\right)\right)^{1/2}\wedge \mu^{-1}\right),
>     $$
>     where $C$ is a positive constant.
>
>  Under the more general condition on $\mu_i$, the **second part of Lemma 4.2 remains valid (when $K\ge M$)**, while the **first part becomes weaker**. As a result, the **adversarial regret bound becomes slightly worse** than before:
>
> $$
> \mathcal{O}\left(\sqrt{nmd}\left(\sqrt{\log(d)}+\frac{m^{5/6}}{d^{1/2}}\right)\right).
> $$
>
> The additional factor $\frac{m^{5/6}}{d^{1/2}}$ is negligible when $d \gg m$.
>
> In the **stochastic setting**, the **logarithmic regret guarantee still holds**, though the additive term becomes slightly larger:
>
> $$\mathcal{O}\left(\sum_{i,\Delta_i>0}\frac{\log(n)}{\Delta_i}\right)+\mathcal{O}\left(\frac{1}{\Delta}(m^2d\log(d)+m^{11/3}+md^2)\right)$$
>
> Therefore, our results **still demonstrate that the FTPL algorithm approaches the BOBW guarantee**.
>
> We will include the complete proof of the new version of Lemma 4.2 at the end of our response. Since the derivation of the modified regret bound based on this lemma does not differ significantly from the corresponding parts in the original version of the paper, we do not provide the full derivation here. However, we are happy to include it in a follow-up response if necessary.
>
> In addition, we have provided a new lower bound to demonstrate that the newly introduced $\frac{m^{5/6}}{d^{1/2}}$ term is also unavoidable. Please refer to our response to Reviewer 7BAU (Weakness 2) for further details.
>
> **Question**:
> Regarding the work of Chen, Botao, and Junya Honda that you mentioned, please allow us to reiterate a few points:
>
> 1. Our work is still the first to show that FTPL approaches the BOBW guarantee. We would like to emphasize that our result in the stochastic setting should not be overlooked — it is not a straightforward extension (see Lemma 4.5 for details).
>
> 2. Although our result in the adversarial setting is slightly weaker, as an early work studying the BOBW problem for FTPL in the semi-bandit setting, we theoretically demonstrate the limitations of current techniques by providing a matching lower bound. This contribution helps clear the path for future progress.
>
> 3. Compared to Lemma 4.1, the proof of Lemma 4.2 plays a more critical role in our analysis. Our careful treatment of the ratio $H_{N+k}/H_N$ allows us to avoid relying on the monotonicity condition in the original Lemma 4.1.
>
> >Proof of modified Lemma 4.2: Let $u = \frac{1}{x + \mu}$. By the change of variable in the integral, we obtain
> $$
> H_N=\int_0^{\mu^{-1}} u^{N-2}e^{-Ku^2}\prod_{i=1}^M \left(1-F\left(\frac{1}{u}+\mu_i-\mu \right)\right)du.
> $$
> We now consider a random variable $U$ with probability density function $g(u)$ defined as
> $$
> g(u) = \frac{1}{\Lambda}u^{N-2}e^{-K u^2}\prod_{i=1}^M\left(1-F\left(\frac{1}{u}+\mu_i-\mu \right)\right),\quad u\geq 0,
> $$
> where $\Lambda$ is the normalization constant. With this, we have
> $$
> \frac{H_{N+k}}{H_N} = E[U^k].
> $$
> **Our plan** is to find some $u_0 > 0$ such that $g(u)$ **decays rapidly** after $u_0$, then we expect that the order of $E[U^k]$ is roughly $u_0^k$.
> **We first show the first result.** The upper bound of $\mu^{-k}$ is obvious because $(x+\mu)^{-N-k}\leq \mu^{-k}(x+\mu)^{-N}$ and hence, in the following we assume that $\mu^{-1}\ge C_{N}'\left(\frac{M}{K}\right)^{1/3}$, where $C_{N}'=8\sqrt{N-2}$.  Let $\ell(u)=\log g(u)$ and $y_i=\frac{1}{u}+\mu_i-\mu$ for all $1\leq i\leq M$, then when $u\ge 0$,
>         $$
>         \frac{d \ell}{d u}(u)=\frac{N-2}{u}-2Ku+2\sum_{i=1}^M\frac{1_{\{y_i\ge 0\}}}{u^2y_i^3\left(e^{\frac{1}{y_i^2}}-1\right)}.
>         $$
> Then we will show that when $u\ge u_0:=C_{N}'\left(\frac{M}{K}\right)^{1/3}/2$, $\frac{d \ell}{d u}(u)\leq -Ku/2$. Intuitively, this suggests that for $u \ge u_0$, $U$ exhibits a Gaussian tail. Then, by applying Lemma G.10 (which we include at the end), we can conclude that
> $$
> E[U^k] \leq u_0^k + E[U^k 1_{\{\mu^{-1} \ge U \ge u_0\}}] \leq (1 + 2k!!) u_0^k.
> $$
> To show that $\frac{d \ell}{d u}(u)\leq -Ku/2$, it suffices to note that $\sup_{x\ge0}\frac{1}{x^3\left(e^{\frac{1}{x^2}}-1\right)}<1$, then we have
>         $$
>         \frac{d\ell}{du}(u)\leq\frac{N-2}{u}-2Ku+\frac{2M}{u^2}.
>         $$
>         When $u\ge u_0$, clearly, since $C_N$ is large enough, we have
>         $$
>         \frac{2M}{u^2}\leq \frac{Ku}{2}.
>         $$
>         Also, since $u_0>\sqrt{\frac{N-2}{K}}$, when $u\ge u_0$, we have
>         $\frac{N-2}{u}\leq Ku.$
> Therefore, when $u\ge u_0$, $$\frac{d\ell}{du}(u)\leq -Ku/2.$$ **For the second part ($N=3$ and $k=1$)**, let $$u_0=\left(\frac{M}{K}\log\left(\frac{K}{M}+1\right)\right)^{1/2}.$$ When $K\leq 32M$, by the first part, the result holds clearly when $C$ is large enough. Hence we then assume that $K>32M$. Then $u_0<1/3$. Similarly, in the following, we also assume that $\mu^{-1}\ge C'u_0$, where $C'>3$ is a large constant to be chosen and is not depending on $K$ and $M$.
> For all $s\ge t\ge 1$, we have
>         $$
>         \frac{g(su_0)}{g(tu_0)}=\frac{s}{t}\mathrm{e}^{-Ku_0^2(s^2-t^2)}\prod_{i=1}^M\frac{1-F\left(\frac{1}{su_0}+\mu_i-\mu\right)}{1-F\left(\frac{1}{tu_0}+\mu_i-\mu\right)}.
>         $$
>         If $1\leq t\leq 3$, then $tu_0<1$. Hence, by Lemma G.7 (which we include at the end), we have$$
>             \frac{g(su_0)}{g(tu_0)}\leq \frac{s}{t}\mathrm{e}^{-Ku_0^2(s^2-t^2)}\left(\frac{8}{t^2u_0^2}\right)^M\leq s\mathrm{e}^{-Ku_0^2(s^2-t^2)}\left(\frac{8}{u_0^2}\right)^M
>  $$
>         Then on the one hand, for all $t\in[1,2]$, we have
>         $$
>          \frac{g(3u_0)}{g(tu_0)}\leq 3\mathrm{e}^{-5Ku_0^2}\left(\frac{8}{u_0^2}\right)^M\leq 3\mathrm{e}^{-5M\log\left(\frac{K}{M}+1\right)}\left(\frac{8K}{M}\right)^M<1,
>         $$
>         where we used the definition of $u_0$ in the second inequality and $\frac{K}{M}>8$ in the last inequality.
>         Then since $C'\ge 2$,
>         $$
>         1\ge \int_{u_0}^{\mu^{-1}}g(u) du\ge \int_{u_0}^{2u_0}g(u) du\ge u_0 g(3u_0),
>         $$
>         which implies that $g(3u_0)\leq u_0^{-1}.$
> On the other hand, for all $s\ge 3$, we have
>         $$
>         \frac{g(su_0)}{g(3u_0)}\leq s\mathrm{e}^{-Ku_0^2(s^2-9)}\left(\frac{8}{u_0^2}\right)^M\leq\mathrm{e}^{\log (s)-M\log\left(\frac{K}{M}+1\right)(s^2-9)+M\log(8)+M\log\left(\frac{K}{M}\right)}.
>         $$
>         Then since $K>32M\ge 32$, one can find $C''>3$ that is not depending on $K$ and $M$ and large enough (one can then pick an $C'$ larger than $C''$) such that for all $s\ge C''$, we have
>         $$
>         \frac{g(su_0)}{g(3u_0)}\leq\mathrm{e}^{-\frac{M}{2}\log\left(\frac{K}{M}+1\right)s^2}.
>         $$
>         Therefore,
>         $$
> \int_{C''}^{\frac{\mu^{-1}}{u_0}}s g(su_0)ds\leq g(3u_0)\int_{C''}^{+\infty}s\mathrm{e}^{-M\log\left(\frac{K}{M}+1\right)s^2/2} ds\leq u_0^{-1}\int_{C''}^{+\infty}s\mathrm{e}^{-M\log\left(\frac{K}{M}+1\right)s^2/2} ds,
>         $$
>         which equals
>         $$
> u_0^{-1}\left(M\log\left(\frac{K}{M}+1\right)\right)^{-1}\mathrm{e}^{-C''^2M\log\left(\frac{K}{M}+1\right)/2}\leq u_0^{-1}.
>         $$
>         Then we have
>         $$
>             E[U]\leq C''u_0+\int_{C''u_0}^{\mu^{-1}} ug(u)du=C''u_0+u_0^2\int_{C''}^{\frac{\mu^{-1}}{u_0}} sg(su_0)ds\leq (C''+1)u_0.
>         $$
>
> Finally, we include the auxiliary lemmas:
> >Lemma G.7: For all $\mu\in R$ and $y\ge x>0$, if $y\ge 1$, then we have
>     $$
>     \frac{1-F(x+\mu)}{1-F(y+\mu)}\leq 8y^2.
>     $$
> > Proof: If $x+\mu\leq y$, then
>         $$
>         \frac{1-F(x+\mu)}{1-F(y+\mu)}\leq\frac{1}{1-F(y+\mu)}.
>         $$
>         When $y+\mu\leq 0$, the right hand equals $1$ and is clearly less than $8y^2$. Otherwise, $y+\mu>0$ and then it suffices to note that
>         $$
>         1-F(y+\mu)=1-e^{-(y+\mu)^{-2}}\ge 1-e^{-y^{-2}/4}\ge y^{-2}/8,
>         $$
>         where we used $y+\mu\leq y+x+\mu\leq 2y$ in the first inequality and $1-e^{-x}\ge x/2$ when $0\leq x\leq 1$ in the second inequality.
> If $x+\mu>y$, then $y+\mu\ge x+\mu>y\ge 1$ and similarly,
>         $$
>         1-F(y+\mu)=1-e^{-(y+\mu)^{-2}}\ge(y+\mu)^{-2}/2.
>         $$
>         Also, since for all $x\ge 0$, $1-e^{-x}\leq x$, we then have
>         $$
>         1-F(x+\mu)=1-e^{-(x+\mu)^{-2}}\leq(x+\mu)^{-2}.
>         $$
>         Therefore,
>         $$
>         \frac{1-F(x+\mu)}{1-F(y+\mu)}\leq 2\left(\frac{y+\mu}{x+\mu}\right)^2<2\left(\frac{y+y-x}{y}\right)^2\leq 8\leq 8y^2.
>         $$
>
> >Lemma G.10: For any random variable $ X $, let its density be denoted by $ f(x) $. If there exist $\mu\ge2\sigma > 0$ and $\mu'>\mu$ such that
> $$
> \frac{d\log f}{dx}(x) \leq -x/\sigma^2, \quad \forall \mu'\ge x \ge \mu,
> $$
> then for all $k\in\mathcal{N}^+$, if $\mu'\ge2\mu$, we have
> $$
> E[X^k  1_{\{\mu'\ge X \ge \mu\}}]\leq 2k!!\cdot\mu^k.
> $$
>
> Lemma G.10 can be viewed as a comparison result: the expectation is maximized when equality holds, in which case $X$ becomes a truncated Gaussian, and the expectation can be computed inductively. Due to space limitations, we omit the full proof but can provide it in a follow-up if needed.

---

> > ### Comment · Reviewer_1er3 · 2025-08-02
> >
> > I greatly appreciate the authors’ efforts in addressing my concerns.
> >
> > I carefully reviewed the rebuttal, including the responses to other reviewers. Overall, the revised arguments appear correct, although some parts were a bit difficult to follow due to omitted details due to space limit.
> > While I was able to follow most of the discussion, one point remains unclear:
> >
> > > Therefore, $$ \int_{C''}^{\frac{\mu^{-1}}{u_0}}s g(su_0)ds\leq g(3u_0)\int_{C''}^{+\infty}s\mathrm{e}^{-M\log\left(\frac{K}{M}+1\right)s^2/2} ds\leq u_0^{-1}\int_{C''}^{+\infty}s\mathrm{e}^{-M\log\left(\frac{K}{M}+1\right)s^2/2} ds, $$ which equals $$ u_0^{-1}\left(M\log\left(\frac{K}{M}+1\right)\right)^{-1}\mathrm{e}^{-C''^2M\log\left(\frac{K}{M}+1\right)/2}\leq u_0^{-1}. $$
> >
> > Q. Could you clarify how the second inequality leads to the stated upper bound (the “which equals” part)? A more detailed explanation of this step would be helpful.
> >
> > ---
> >
> > Additional comments:
> >
> > I also noticed a small issue with Lemma 4.3. The expression $\hat{\underline{L}}_{t,i}$ can be negative when $\sigma_i(\hat{L}_t) \leq m$, where the integral around Line 552-553 may not be well-defined.
> > Although this does not seem to affect the actual proof, since the first term is only invoked when $\sigma_i>m$, the lemma as currently stated is technically incorrect and should be revised accordingly.

---

> > > ### Author Response · Authors · 2025-08-02
> > >
> > > We sincerely thank you for carefully reading our responses and for your kind recognition of our effort, which is greatly encouraging to us.
> > >
> > > - First, we appreciate your new question and provide the following detailed explanation: The step "which equals" corresponds to the result of computing the integral in the previous line. We note that the integrand (with respect to $s$), namely  $
> > > u\_0^{-1}s \mathrm{e}^{-M\log\left(\frac{K}{M}+1\right)s^2/2},
> > > $
> > > admits an antiderivative:
> > > $$
> > > -u\_0^{-1}\left(M\log\left(\frac{K}{M}+1\right)\right)^{-1}\mathrm{e}^{-M\log\left(\frac{K}{M}+1\right)s^2/2}.
> > > $$
> > > Therefore, the integral equals
> > > $$
> > > -u\_0^{-1}\left(M\log\left(\frac{K}{M}+1\right)\right)^{-1}\mathrm{e}^{-M\log\left(\frac{K}{M}+1\right)s^2/2} \Big|\_{C''}^{+\infty} = u\_0^{-1}\left(M\log\left(\frac{K}{M}+1\right)\right)^{-1} \mathrm{e}^{-C''^2 M\log\left(\frac{K}{M}+1\right)/2}.
> > > $$
> > > Since $K \ge M\ge 1$ and $C'' > 3$, this result is less than $u\_0^{-1}$. We will include this explanation in the revised version.
> > >
> > > - Second, thank you for pointing out that some parts were a bit difficult to follow due to omitted details caused by space limitations. We acknowledge this and believe that the main difficulty likely lies in the second part of the proof. Therefore, we would like to provide here an additional intuitive explanation for that part.
> > >
> > >   As stated at the beginning of the proof, our plan is to find some $u\_0 > 0$ such that $g(u)$ decays rapidly after $u\_0$, and then we expect that the order of $E[U^k]$ is roughly $u\_0^k$. Guided by this idea, the first step in our reasoning was to compute the ratio $\frac{g(su\_0)}{g(u\_0)}$ for $s \ge 1$, and we found that for a suitable choice of $u\_0$ (as done in the formal proof), this ratio decays at a rate comparable to the Gaussian tail, which is precisely what we hoped for.
> > >
> > >
> > >    However, the challenge is that we also need to ensure that $g(u\_0)$ is not too large in order to concretely control the decay rate of $g(su\_0)$. To illustrate this, we used a simple fact: if we choose $u\_0$ large enough so that $g(u)$ starts decreasing after $\frac{u\_0}{3}$, then the values of $g(u)$ over the interval $[\frac{u\_0}{3}, \frac{2u\_0}{3}]$ should all be larger than $g(u\_0)$. Since $g(u)$ is a probability density function, we then have
> > >   $$1 \ge \int\_{\frac{u\_0}{3}}^{\frac{2u\_0}{3}} g(u)\,du \ge g(u\_0) \cdot \frac{u\_0}{3},$$
> > >
> > >   which naturally gives an upper bound on $g(u\_0)$. (Note that in the proof we scaled $u\_0$ to $3u\_0$ for convenience of presentation.) We will elaborate further on this point in the revised version.
> > >
> > > - Third, regarding the Additional comments, thank you for your careful reading. In fact, we have already defined in the *Notation* section (line 111) that
> > > $$
> > > \underline{\lambda}\_i := \left(
> > > \lambda\_i - \max\limits\_{\mathcal{I}, |\mathcal{I}|<m} \min\limits\_{j \notin \mathcal{I}} \lambda\_j
> > > \right)^+\ge 0,
> > > $$
> > > so when $\sigma\_i(\hat{L}\_t) \leq m$, we indeed have $\underline{\hat{L}}\_{t,i} = 0$, which avoids the issue you mentioned. We will make sure to emphasize this point more clearly in the revised version.
> > >
> > > We once again thank you for your careful and thorough review — it has been very helpful to us. If you have any further questions, we would be more than happy to discuss them.

---

> > > > ### Comment · Reviewer_1er3 · 2025-08-02
> > > >
> > > > Thank you for your quick and detailed explanations to my questions, including the intuitions behind them!
> > > >
> > > > After reading the clarification on the first part, I realized that I had mistakenly interpreted “which equals” as $\iff$, whereas it was simply $=$ connecting two lines.
> > > > This is now clear. I also appreciate the pointer to the definition of the underline; I had forgotten that the authors introduced a clipping function. Sorry for the confusion.
> > > >
> > > > ---
> > > >
> > > > As far as I understand, the modified adversarial result is
> > > > $$
> > > > O(\sqrt{nmd\log d} + m^{4/3}\sqrt{n}).
> > > > $$
> > > > Therefore, when  $(d \log d)^{\frac{3}{5}} \leq m \leq d/2$, the adversarial regret becomes worse than expected. However, as the authors noted, it cannot be improved within the current proof framework.
> > > >
> > > > Nevertheless, this work presents the first BOBW guarantee for FTPL under the combinatorial semi-bandit setting, where the adversarial regret is at least an improvement over the previous FTPL bound of $O(m\sqrt{nd\log (d/m)})$, while also achieving logarithmic stochastic regret.
> > > > Although I expect that the adversarial regret can be improved further, doing so would likely require a different type of analysis or even a different choice of perturbations, which would be beyond the scope of this paper.
> > > >
> > > > Although the current rebuttal policy makes it difficult to fully verify every detail, I think the modified proof is convincing. Therefore, I will raise my score to 4, while lowering my confidence to 3.

---

> > > > > ### Author Response · Authors · 2025-08-02
> > > > > **Thank you!**
> > > > >
> > > > > We are deeply appreciative of your insightful and beneficial comments on our manuscript. We will incorporate the suggestions in the revised version.

---

### Official Review · Reviewer_JPYF · 2025-07-02

**Clarity:** 2
**Significance:** 2
**Originality:** 2
**Rating:** 4
**Confidence:** 3

**Summary:**

This paper introduces and analyzes the Follow-the-Perturbed-Leader (FTPL) policy with Frechet perturbations for the m-set semi-bandit problem, which aims overcome the computational complexities of the traditional Follow-the-Regularized-Leader (FTRL) policies. The main contribution is achieving Best-of-Both-Worlds (BOBW) regret guarantees, making it the first FTPL algorithm to approach this performance in the m-set semi-bandit setting when $m\leq d/2$. Specifically, it demonstrates a near-optimal regret in the adversarial setting and logarithmic regret in the stochastic setting. The authors also establish a lower bound and indicate that the logarithmic factor $(log(d))$ in the adversarial regret is inherent to their analytical approach, and eliminating it would require a fundamentally different method. Furthermore, the work advances theoretical analysis by extending the standard FTRL framework to m-set semi-bandits, a setting where the convex hull of the action set lacks interior points, thereby simplifying previous proofs. The effectiveness of the proposed FTPL algorithm is also supported by empirical validation against various established baselines.

**Questions:**

1) Could you provide a more in-depth discussion of the specific technical challenges that prevent the removal of the $log(d)$ factor in the adversarial regret bound? It is only noted that obtaining a tighter bound is not possible through bounding the term-wise ratio; and instead, one must analyze the ratio of the full summations directly, which is substantially more challenging. A clearer understanding of this analytical hurdle would be valuable.

2) What are the specific hurdles encountered when attempting to extend the FTPL's BOBW guarantee to the $m > d/2$ regime. It is indicated that FTRL algorithms have achieved BOBW guarantees in this setting, but whether FTPL can match this remains an open problem.

3) What are the implications if there are more than $m$ arms with zero suboptimality gap? How would the analytical framework (e.g., the regret decomposition for the stochastic setting) break down or degrade if this assumption were relaxed?

**Ethical Concerns:**

["NO or VERY MINOR ethics concerns only"]

**Final Justification:**

I am keeping my score, which remains positive.

My concern regarding the unavoidable log(d) factor in the adversarial regret bound has been effectively addressed. The strengthening of Lemmas and related lower bounds clearly demonstrate that this factor is inherent to your current analytical approach. The discussions confirmed the complexity and rigor of your theoretical analysis for FTPL in the m-set semi-bandit setting, especially concerning the arm-selection probabilities and extending the FTRL framework where the convex hull of the action set lacks interior points. The clarified regret decomposition also enhanced confidence in the proof's integrity. The comprehensive empirical evaluation remains a strong aspect, supporting the practical efficacy of your proposed method across stochastic and adversarial settings.

The assumption of a unique optimal action in the stochastic setting persists. While your rebuttal suggests FTPL can achieve logarithmic regret without this assumption, it remains an unproven conjecture within the paper and is deferred as future work. The theoretical guarantee in the paper still relies on this specific assumption. The paper's Best-of-Both-Worlds (BOBW) guarantee currently applies when m ≤ d/2. Extending FTPL to the m > d/2 regime remains an acknowledged open problem, noted as an intriguing and challenging future direction.

My decision is primarily driven by the theoretical contribution of presenting the first FTPL algorithm with Best-of-Both-Worlds guarantees in the m-set semi-bandit setting

**Limitations:**

yes

**Paper Formatting Concerns:**

no major issues

**Quality:**

3

**Strengths And Weaknesses:**

# Strengths

1) The FTPL policy achieves optimal BOBW regret in both adversarial and stochastic settings, which is an advantage for uncertain environments.

2) FTPL is computationally efficient as it directly selects $m$ arms with smallest perturbed losses, avoiding complex arm-selection probability computations required by FTRL. It uses geometric resampling for unbiased estimators.

3) The paper extends the standard FTRL analysis framework to m-set semi-bandits where the action set's convex hull lacks interior points, simplifying prior proofs by Honda et al.. It also provides a lower bound showing that the log(d) factor in adversarial regret is inherent to their approach.

4) The paper provides detailed empirical validation by comparing FTPL against five baselines (COMBUCB, Thompson Sampling, EXP2, LogBarrier, and a hybrid FTRL) across both stochastic and adversarial environments, with specified parameters, loss generation, and repetitions.

# Weaknesses

1) The theoretical analysis for the stochastic setting (Theorem 3.2) assumes the uniqueness of the optimal action, which might not hold universally in practice.

2) While its application seems novel, FTPL itself is an existing algorithm. The contribution is an extension and refinement of prior FTRL/FTPL analysis, rather than a new algorithmic approach.

3) The FTPL method's adversarial regret includes an unavoidable $log(d)$ factor, preventing it from strictly matching the minimax optimal bound.

4) The BOBW guarantees are only applicable when $m \leq d/2$. Extending these results to the $m > d/2$ regime remains an open problem.

---

> ### Author Rebuttal · Authors · 2025-07-30
>
> Thank you for your valuable review and constructive suggestions. Regarding the weaknesses and questions, we provide the following detailed responses:
>
> **Weakness 1**: Though the uniqueness of the optimal action is a common assumption in BOBW problems  [Zimmert and Seldin, 2019, Zimmert et al., 2019, Honda et al., 2023], we would like to emphasize that this is primarily for technical convenience (see our response to Question 3 for further discussion). Intuitively, the case where the optimal action is not unique is actually easier for the learner. In the MAB setting, some classical BOBW algorithms based on FTRL have already been shown to achieve logarithmic regret even without the uniqueness assumption, though such results require more delicate analysis (e.g., Parameter-Free Multi-Armed Bandit Algorithms with Hybrid Data-Dependent Regret Bounds, Shinji Ito; Improved Best-of-Both-Worlds Guarantees for Multi-Armed Bandits: FTRL with General Regularizers and Multiple Optimal Arms, Tiancheng Jin, Junyan Liu, and Haipeng Luo).
>
> Our additional experiments (We're sorry, but due to the new NeurIPS policy, we are temporarily unable to upload images.) show that even when the optimal action is not unique, FTPL still achieves logarithmic regret in the stochastic setting. This supports our conjecture that the assumption is not essential. We view this as a promising direction for future work and will include a discussion in the revised version. Thank you for bringing this up.
>
> **Weakness 2**: We acknowledge that our work does not introduce algorithmic innovations. Our main contribution lies in the theoretical analysis of the FTPL algorithm in the semi-bandit setting, which is technically challenging. Specifically:
>
> 1. Our analysis of the ratio $\frac{V_{i,4}}{V_{i,3}}$—where both the numerator and denominator involve complex integrals and summations—is nontrivial. By carefully analyzing the ratio of corresponding terms in the numerator and denominator (including both upper and lower bounds), we obtain a nontrivial result for the first time, and further demonstrate the theoretical limitations of this approach.
>
> 2. We would also like to emphasize that our result in the stochastic setting should not be overlooked — it is not a straightforward extension (see Lemma 4.5 for details).
>
>
> **Weakness 3**: We demonstrate that this is difficult to avoid within our proof framework by establishing a lower bound. Any further improvement would likely require adopting a different and more challenging line of analysis.
>
>
>
> **Weakness 4**: We believe it is likely that, without modifying the algorithm, FTPL with a Fréchet distribution of shape parameter 2 cannot achieve the optimal performance in the adversarial setting when $m \ge d/2$, and therefore fails to attain the BOBW guarantee. Please refer to our response to Question 2 for details.
>
>
> **Question 1**:
> We provide a detailed explanation of this issue. According to our analysis in Section 4.1, the stability term in round $t$ can be bounded by
> $$
> \eta_t \sum_{i=1}^d \frac{V_{i,4}(\lambda)}{V_{i,3}(\lambda)}.
> $$
> The **main challenge** lies in the fact that $V_{i,N}(\lambda) = \sum_{I,\ |I|<m,\ i \notin I} V_{i,N}^{I}(\lambda)$ is a sum over a collection of complicated integrals, which makes it difficult to directly compute $\frac{V_{i,4}(\lambda)}{V_{i,3}(\lambda)}$. A natural idea is to use the upper bound:
> $$
> \frac{V_{i,4}(\lambda)}{V_{i,3}(\lambda)} \leq \sup_{I,\ |I|<m,\ i \notin I} \frac{V_{i,4}^{I}(\lambda)}{V_{i,3}^{I}(\lambda)}.
> $$
> However, our lower bound shows that when $\sigma_i(\hat{L}_t) > 2m$, we have
> $$
> \sup\_{\lambda} \sup\_{I,\ |I|<m,\ i \notin I} \frac{V\_{i,4}^{I}(\lambda)}{V\_{i,3}^{I}(\lambda)} \ge \sqrt{\frac{m \log(d)}{\sigma\_i(\hat{L}\_t) - m}}.
> $$
> This implies that, under this approach, the upper bound of the stability term is at least
> $$
> \eta_t\sum_{i:\ \sigma_i(\hat{L}_t) > 2m} \sqrt{\frac{m \log(d)}{\sigma_i(\hat{L}_t) - m}} = \Omega\left(\sqrt{\frac{m d \log(d)}{t}}\right).
> $$
>
> **Question 2**: According to the result of "Beating Stochastic and Adversarial Semi-bandits Optimally and Simultaneously" by Julian Zimmert, Haipeng Luo, and Chen-Yu Wei (Theorem 3), when $ m > d/2 $, the optimal regret bound in the adversarial setting is
> $$
> O\left((d - m)\sqrt{\log\left(\frac{d}{d - m} \right)n}\right).
> $$
> However, if we choose the Fréchet distribution with shape parameter 2 as the perturbation distribution, the resulting penalty term remains
> $$
> O(\sqrt{nmd}),
> $$
> **regardless of the relationship between $ m $ and $ d $.** This stems from a simple fact: consider $ X_1, \ldots, X_d $ as i.i.d. random variables drawn from the Fréchet distribution with shape parameter 2. Then, the expectation of the sum of the top $ m $ among $ X_1, \ldots, X_d $ scales as $ \sqrt{dm} $.
>
> However, this is too large when compared to $(d - m)\sqrt{\log\left(\frac{d}{d - m} \right)}$ in the optimal bound in the case where $ m > d/2 $ (when $m=d-1$, for example). Therefore, we believe that, without modifying the algorithm, FTPL with a Fréchet distribution (shape parameter 2) is unlikely to achieve the optimal regret in the adversarial setting when $ m \ge d/2 $.
>
> It is important to emphasize that the algorithm in "Beating Stochastic and Adversarial Semi-bandits Optimally and Simultaneously" that achieves the BOBW guarantee even when $ m > d/2 $ is based on FTRL with a hybrid regularizer. The hybrid regularizer used there is relatively complex and thus can adapt to various values of $ m $ and $ d $. To modify FTPL to handle more general cases, it may similarly require more sophisticated perturbation distributions.
>
>
> **Question 3**:
> The difficulty of the case where the optimal action is not unique lies in the following aspects:
>
> 1. The lower bound used in the self-bounding constraint technique,
> $$
> Reg_n \ge \Delta \sum_{t=1}^n E\left[
> \sum_{i=m+1}^d w_{t,i}
> \right],
> $$
> no longer holds. This is because, in this case, the $m$-th and $(m+1)$-th arms may have the same expected loss, and both can potentially appear in some optimal action sets. As a result, selecting the $(m+1)$-th arm does not necessarily incur at least $\Delta$ regret. Therefore, a more refined characterization of the lower bound is required, which becomes significantly more cumbersome.
>
> 2. Once a new lower bound is obtained, we must correspondingly strengthen Lemma 4.5 to match the upper and lower bounds. This requires more sophisticated analytical techniques from FTRL algorithms (see "Parameter-Free Multi-Armed Bandit Algorithms with Hybrid Data-Dependent Regret Bounds", Shinji Ito; and "Improved Best-of-Both-Worlds Guarantees for Multi-Armed Bandits: FTRL with General Regularizers and Multiple Optimal Arms", Tiancheng Jin, Junyan Liu, and Haipeng Luo).

---

> > ### Comment · Reviewer_JPYF · 2025-08-05
> >
> > I appreciate the authors' rebuttal and have some follow up questions.
> >
> > - In response to Weakness 1 and Question 3, you state that FTPL can still achieve logarithmic regret in the stochastic setting even when the optimal action is not unique, viewing this as a promising direction for future work. This is a significant claim that directly addresses a major theoretical assumption. Is it an observation over the numerical experiments or is there any theoretical basis for it? If so, could you provide a brief sketch or more technical intuition on how the theoretical analysis (e.g., the regret decomposition in Lemma 3.3 or the bounds in Theorem 3.2) is adapted to guarantee this logarithmic regret in the absence of a unique optimal action? How are the suboptimality gaps defined and handled in this more general scenario, and what implications does this have for the $\Delta$ term in the regret bound?
> >
> > - Your rebuttal for Weakness 3 and Question 1 clearly explains that the $\log(d)$ factor is inherent to your current proof framework due to the tightness of Lemma 4.2 and the complexity of analyzing the ratio of full summations directly. While this clarifies the limitations of your methodology, does the current state of research suggest that this $\log(d)$ factor is inherent to any FTPL algorithm aiming for Best-of-Both-Worlds (BOBW) guarantees in the m-set semi-bandit setting, or is there a theoretical possibility that a fundamentally different FTPL approach could achieve the minimax optimal regret?
> >
> > - Regarding Weakness 4 and Question 2, you state that FTPL with Fréchet perturbation (shape parameter 2) cannot achieve optimal performance when $m \ge d/2$, and suggest that "more sophisticated perturbation distributions" might be needed. Could you elaborate on what properties these "more sophisticated" distributions would need to possess to overcome the current limitations for $m \ge d/2$? Is there any intuitive explanation as to why the standard Fréchet perturbation specifically fails in this regime compared to the $m \le d/2$ case?

---

> > > ### Author Response · Authors · 2025-08-06
> > >
> > > We sincerely thank you for your response. Regarding your question about the open problem discussed in our paper, although it goes beyond the scope of this work, we are more than happy to engage in the discussion. We provide the following explanation:
> > >
> > > First, regarding the first question, this remains our conjecture—we believe it is a promising direction for future work, but no theoretical guarantee has been established yet. As we mentioned in our previous response, beyond the numerical evidence, we see two main intuitive reasons supporting this conjecture:
> > >
> > >  1. Intuitively, the setting with non-unique optimal actions should be easier, since the player has a higher chance of selecting an optimal action.
> > >
> > >  2. In the literature, many FTRL-based algorithms achieving BOBW in bandit problems have also been shown to attain logarithmic regret in the non-unique optimal action setting, without requiring any modification.
> > >
> > >
> > > Second, regarding the second question, this is not the case. We tend to believe that the $\log(d)$ factor arises from the limitations of our current proof technique rather than an inherent limitation of the FTPL algorithm itself. As we emphasized in our previous response, FTPL with a Fréchet distribution (shape parameter 2) yields a penalty term of exactly $\sqrt{nmd}$, which matches the optimal regret bound.
> > >
> > > Moreover, based on studies of FTRL algorithms, it has been observed that when the regularizer is chosen such that the resulting penalty term matches the optimal regret bound, the stability term also often aligns with the optimal rate (see, e.g., Tsallis-INF: An Optimal Algorithm for Stochastic and Adversarial Bandits, Julian Zimmert and Yevgeny Seldin). Therefore, we are inclined to believe that FTPL with a Fréchet distribution (shape parameter 2) can indeed achieve the optimal regret bound.
> > >
> > > Finally, regarding the third question, we are currently unable to determine how FTPL should be modified in order to achieve optimal performance when $m \ge d/2$—this lies completely beyond the scope of the current paper. In fact, we would like to emphasize that simply changing the perturbation distribution may still not resolve this issue.
> > >
> > > We offer the following intuitive explanation. As analyzed in the main text, the penalty term of the FTPL algorithm is roughly $\sqrt{n}$ times the expectation of the sum of the top $m$ values among $X_1, \ldots, X_d$, where $X_1, \ldots, X_d$ are i.i.d. samples from the perturbation distribution. This quantity clearly increases with $m$—that is, the larger $m$ is, the larger the penalty term becomes.
> > >
> > > However, intuitively, the problem should become easier for the player when $m > d/2$ and $m$ is larger, since the number of available actions becomes smaller. For instance, in the extreme case of $m = d - 1$, selecting $d - 1$ arms per round means there are only $d$ possible actions in total. This seems to contradict the monotonicity of the penalty term with respect to $m$ (a more formal version of this explanation can be found in our earlier response from the perspective of the optimal regret bound).
> > >
> > > Therefore, to handle the case $m > d/2$, a substantial modification of the FTPL algorithm may be necessary—likely beyond simply adjusting the perturbation distribution. This currently appears to be a very challenging problem, but we believe it is a promising and meaningful direction for future research.
> > >
> > > Please allow us to once again thank you for your review and feedback — it has been extremely helpful to us.

---

### Official Review · Reviewer_S1FU · 2025-07-03

**Clarity:** 2
**Significance:** 3
**Originality:** 3
**Rating:** 4
**Confidence:** 3

**Summary:**

This paper studies the BoBW semi-bandit problem. Utilizing and refining recent technical advancements in FTPL with Frechet-type perturbations, this paper derived an FTPL-based algorithm achiving the $\tilde{\mathcal O}(\sqrt{nmd})$ adversarial regret and $\mathcal O(\sum_i \log n \Delta_i^{-1})$ stochastic regret. While the regret is slightly sub-optimal in terms of logarithmic factors, FTPL avoids the hard-to-solve global optimization problem common in FTRL.

**Questions:**

See weaknesses.

**Ethical Concerns:**

["NO or VERY MINOR ethics concerns only"]

**Final Justification:**

I am satisfied with Authors' responses and decide to keep my original evaluation.

**Limitations:**

Yes

**Quality:**

3

**Strengths And Weaknesses:**

Strength:

1. The elegance and efficiency of FTPL is for sure a plus.
2. New analytical tools for FTRL is proposed (i.e., it's not a simple adaptation of recent BoBW FTPL results to semibandits).
3. Numerical illustrations are given.

Weakness:

The paper is written in a super technical way and lacks some justifications even for experts in this area. For example, around Line 169 the claim "which is a stronger result and simplifies the proof compared to those of Honda et al. [2023], Lee et al. [2024]," is not well-elaborated -- while I can get the intuition why this might be simpler as an expert in FTRL, I am not sure whether my understanding is correct.

The notations are also too many to be easily interpreted.

Also, the authors are also strongly encouraged to explain the technical difficulty / insights in proving these results: What leads to this stronger result? Is it because of the FTRL-for-FTPL style analysis? If yes, what do the previous works of Honda et al. and Lee et al. use, and what stopped them from using a similar idea? (Note that I do not mean something like "Line 169 is simply due to an existing observation", but rather suggest the authors to be clear about their contributions.)

Overall, I'm leaning towards an accept because of the good results provided as well as the new contribution to the somehow-scarce FTPL technical toolbox; nevertheless, as I always worked on FTRL instead of FTPL, I'm not super certain how significant this contribution is.

---

> ### Author Rebuttal · Authors · 2025-07-30
>
> We thank you for your review and suggestions. Regarding the question about Lemma 3.3, we provide the following clarifications.
>
> First, our result is stronger in the sense that, compared to previous work by Honda and Lee et al., we obtain an almost identical penalty term, but with a simpler stability term. Specifically, their stability term takes the form
>
> $$
> E\left[\sum\_{t=1}^{n}\langle\hat{\ell}\_t,\phi(\eta_t \hat{L}\_t)-\phi(\eta\_{t+1} \hat{L}\_{t+1})\rangle\right],
> $$
>
> which they further decompose into two parts:
> $$
> E\left[\sum_{t=1}^n
>     \langle\hat{\ell}\_t,\phi(\eta\_t \hat{L}\_t)-\phi(\eta\_{t} \hat{L}\_{t+1})\rangle\right] \quad
>     \text{and}
>     \quad
> E\left[\sum_{t=1}^n
>     \langle\hat{\ell}\_t,\phi(\eta\_{t} \hat{L}\_{t+1})-\phi(\eta\_{t+1} \hat{L}\_{t+1})\rangle\right],
> $$
> and control each part separately. However, under our decomposition, the second term is not needed, simplifying the analysis.
>
> Second, we believe this stronger result indeed stems from adopting an FTRL-for-FTPL-style analytical framework. The Bregman-divergence-based argument is more principled, allowing us to apply the Generalized Pythagoras Identity (Lemma G.2). In contrast, the decomposition in Honda and Lee’s work is more ad hoc, and directly based on the update form
> $$
> A_t = \operatorname{argmin}_{a \in \mathcal{A}} \langle \hat{L}_t - r_t / \eta_t, a \rangle.
> $$
>
> We conjecture there are two main reasons why they did not use the FTRL-for-FTPL framework:
> (1) The regularization function corresponding to FTPL's dual view does not have a closed-form expression, making it hard to analyze;
> (2) Their decomposition is based on the update rule of $A_t$ (i.e., the maximizer of an inner product), which naturally leads to a decomposition where the stability term also takes the form of an inner product. This inner-product form is more convenient for analysis in the context of FTPL algorithms.
>
> As mentioned in Remark 3.1, unlike standard FTRL analyses where the stability term is often approximated by a sum of the form $\eta\_t E[\|\hat{\ell}\_t\|^2\_{\nabla^2 \Phi(-\eta\_t \hat{L}\_t)}]$, this approach does not extend easily to FTPL. This is because $\nabla^2 \Phi(-\eta\_t \hat{L}\_t)$ and $\nabla^2 \Phi(-\eta\_t \hat{L}\_{t+1})$ may differ significantly (we can only control their diagonal entries) — in FTPL, $\nabla^2 \Phi$ is not a diagonal matrix, which is a key difference from typical FTRL algorithms.
>
> In contrast, the inner-product form of the stability term $\langle \hat{\ell}\_t, \phi(\eta\_t \hat{L}\_t) - \phi(\eta\_t \hat{L}\_{t+1}) \rangle$ can be written as
> $$
> \sum\_{i=1}^d \hat{\ell}\_{t,i} \left( \phi\_i(\eta\_t \hat{L}\_t) - \phi_i(\eta\_t \hat{L}\_{t+1}) \right),
> $$
> which, according to Lemma D.2, can be upper bounded by
> $$
> \sum\_{i=1}^d \hat{\ell}\_{t,i} \left( \phi\_i(\eta\_t \hat{L}\_t) - \phi\_i\left(\eta\_t \left(\hat{L}\_t + \hat{\ell}\_{t,i} \cdot e_i\right)\right) \right).
> $$
> In this way, we only need to consider the diagonal entries of $\nabla^2 \Phi$.
>
> However, in any case, we find that a similar and even stronger form of the stability term can also be easily derived using the FTRL-based analysis framework, which further demonstrates the superiority of this framework.

---

> > ### Comment · Reviewer_S1FU · 2025-08-03
> >
> > Thank you for your responses. I encourage the authors to incorporate part of their responses in the revision.

---

> > > ### Author Response · Authors · 2025-08-03
> > > **Thank you!**
> > >
> > > We would like to express our sincere gratitude again for your constructive and valuable comments on our paper. We will incorporate the suggestions (especially add more explanations about Lemma 3.3) in the revised version.

---

### Official Review · Reviewer_7BAU · 2025-07-03

**Clarity:** 3
**Significance:** 2
**Originality:** 2
**Rating:** 4
**Confidence:** 4

**Summary:**

The authors introduced Follow-the-Perturbed-Leader-type algorithm w/ Fréchet-distributed perturbations for the $m$-set combinatorial semi-bandit problem, where a learner selects exactly $m$ out of $d$ arms per round. It is the first to achieve Best-of-Both-Worlds guarantees in this setting: near-optimal $\mathcal{O}(\sqrt{n m d \log d})$ regret under adversarial rewards and logarithmic $\mathcal{O}(\sum_{i:\Delta_i>0} \frac{\log n}{\Delta_i})$ regret for stochastic bandits. FTPL avoids the computational expense of prior FTRL-based BOBW methods by replacing per-step optimization with simple perturbed loss minimization. The analysis reveals the $\log d$ factor is unavoidable via their proof techniques, establishing a "fundamental limit of the *proof strategy*".

**Questions:**

- This paper is self-contained, especially regarding technical details. It is also desirable to have a self-contained high-level illustration of the intuition behind the "Fréchet distribution with shape parameter 2" that enables the BOBW guarantee, right?

**Ethical Concerns:**

["NO or VERY MINOR ethics concerns only"]

**Final Justification:**

Based on the rebuttal, this submission will be a mathematically correct one, which adapts well-established technical toolkits developed by Honda et al. 2023 from bandits to semi-bandits. Therefore, my final decision is borderline accept.

**Limitations:**

Yes.

**Paper Formatting Concerns:**

N/A.

**Quality:**

2

**Strengths And Weaknesses:**

### Strengths

- The high-level idea of using geometric resampling to tackle with the lack of full-information feedback aligns with the intuition in the semi-bandit setting and the regret decomposition in Lemma 3.4 is novel can correct.
- The implication that FTPL without complex tweaks can achieve the best-of-both-worlds guarantee for semi-bandits is a clear and desirable.
- The "lower bound for the proof technique" part (Lemma C.5) is a interesting contribution, which is seldom seen of its kind in the literature.

### Weaknesses

- The last clause from Line 191 to Line 192 is interesting, but its proof from Line 502 to Line 503 is not clear. To be frank, if the authors consider this part as a trivial argument, please give a more clear explanation. However, as far as I can tell, this part is either highly non-trivial or even **incorrect**.
  - **I'm willing to reconsider my evaluation if the authors can convince me that this part is not wrong**.
- Lemma C.5 deserves a more intuitive discussion in the Appendix, as it is a key counterpart of the main result.

---

> ### Author Rebuttal · Authors · 2025-07-30
>
> Thank you for your valuable review and constructive suggestions. Regarding the weaknesses and questions, we provide the following detailed responses:
>
> **Weakness 1**:
> We must acknowledge our oversight: the second half of **Lemma 4.1** was indeed incorrect. To address this issue, we have **strengthened Lemma 4.2** by **removing the assumption $\mu_i \leq \mu$**, which eliminates the need for the monotonicity argument used in the original Lemma 4.1.
>
> >Lemma 4.2 (**modified**):  For all $\mu\ge0$, $K\ge 1$, $ N\ge 3$ and $M\ge 1$, let $\mu_i\in R$ for all $1\leq i\leq M$, and define
>     $$
>     H_{N}=\int_0^{+\infty} (x+\mu)^{-N}e^{-\frac{K}{(x+\mu)^2}}\prod_{i=1}^M\left(1-F(x+\mu_i)\right) d x.
>     $$
>     For all $k\in\mathcal{N}^+$, we have
>     $$
>     \frac{H_{N+k}}{H_N}\leq C_{N,k}\left(\left(\frac{M}{K}\right)^{k/3}\wedge \mu^{-k}\right),
>     $$
>     where $C_{N,k}$ is a positive constant only depending on $N$ and $k$. Furthermore, if $K\ge M$, then we have
>     $$
>     \frac{H_{4}}{H_3}\leq C\left(\left(\frac{M}{K}\log\left(\frac{K}{M}+1\right)\right)^{1/2}\wedge \mu^{-1}\right),
>     $$
>     where $C$ is a positive constant.
>
>  Under the more general condition on $\mu_i$, the **second part of Lemma 4.2 remains valid (when $K\ge M$)**, while the **first part becomes weaker**. As a result, the **adversarial regret bound becomes slightly worse** than before:
>
> $$
> \mathcal{O}\left(\sqrt{nmd}\left(\sqrt{\log(d)}+\frac{m^{5/6}}{d^{1/2}}\right)\right).
> $$
>
> The additional factor $\frac{m^{5/6}}{d^{1/2}}$ is negligible when $d \gg m$.
>
> In the **stochastic setting**, the **logarithmic regret guarantee still holds**, though the additive term becomes slightly larger:
>
> $$\mathcal{O}\left(\sum_{i,\Delta_i>0}\frac{\log(n)}{\Delta_i}\right)+\mathcal{O}\left(\frac{1}{\Delta}(m^2d\log(d)+m^{11/3}+md^2)\right)$$
>
> Therefore, our results **still demonstrate that the FTPL algorithm approaches the BOBW guarantee**.
>
> ## Sorry for the inconvenience. Due to space limitations, we leave the proof of the revised Lemma 4.2 in our response to Reviewer 1er3.
> Since the derivation of the modified regret bound based on this lemma does not differ significantly from the corresponding parts in the original version of the paper, we do not provide the full derivation here. However, we are happy to include it in a follow-up response if necessary.
>
>
> **Weakness 2**:
> We appreciate your interest in the lower bound part of our Lemma C.5. We acknowledge that the current proof of the lower bound may be difficult to follow.
>
>
> An intuitive understanding of this result is that the pdf of $W$, given by $1 - F(x(w) + \mu)$, is monotonically decreasing on $[0,1]$ and clearly exhibits a sharp drop at $e^{-\frac{K}{\mu^2}}$: the pdf is equal to $1$ before this point and then gradually decays afterward. Our calculation shows that most of the density is concentrated in the first half of the interval, and thus the expectation of $(-\log(W))^{1/2}$ is of a larger order than simply substituting $W = e^{-\frac{K}{\mu^2}}$. Thus, by choosing $\mu = -\sqrt{\frac{2K}{\log(K)}}$, the desired lower bound follows.
>
>
> Besides, we would like to emphasize that, due to the appearance of the $\frac{m^{5/6}}{d^{1/2}}$ factor in our new results, we have established a corresponding updated lower bound. This shows that the new Lemma 4.2 is tight, and therefore, the extra factors beyond the optimal rate in the adversarial setting are unavoidable under our approach. To achieve the tightest possible rate, one would have to resort to alternative methods.
>
> >Lemma C.7: For all $\mu\in R$, $M, K\ge 1$, and $ N\ge 3$, define
>     $$
>     R_{N}(\mu)=\int_0^{+\infty} x^{-N}e^{-\frac{K}{x^2}}\left(1-F(x+\mu)\right)^M dx.
>     $$
>     Then there exists $C>0$ such that for all $M\ge 2K$, we have
>     $$
>     \sup_{\mu\in R}\frac{R_{4}(\mu)}{R_3(\mu)}\ge C\left(\frac{M}{K}\right)^{1/3}.
>     $$
>
>
> This new lower bound is originated from the following simple observation: if a random variable $X$ supported on $[0,1]$ has a probability density function $f$, and there exists $x_0 \in (0,1]$ such that $f$ is non-decreasing on $[0,x_0]$, then it holds that $E[X] \geq x_0/4$. This follows from the fact that $P(0 \leq X \leq x_0/2) \leq P(x_0/2 \leq X \leq x_0)$, so $P(0 \leq X \leq x_0/2) \leq 1/2$, which implies $E[X] \geq x_0/4$.
>
> >Proof for Lemma C.7: Let $u = \frac{1}{x}$. By the change of variable in the integral, we obtain
> $$
> R_N(\mu) = \int_0^{+\infty} u^{N-2} e^{-K u^2}  \left(1 - F\left(\frac{1}{u} + \mu \right)\right)^M  du.
> $$
> We now consider a random variable $U$ with probability density function $g(u)$ defined as
> $$
> g(u) = \frac{1}{\Lambda} u^{N-2} e^{-K u^2}\left(1 - F\left(\frac{1}{u} + \mu \right)\right)^M, \quad u \geq 0,
> $$
> where $\Lambda$ is the normalization constant. With this, we have
> $$
> \frac{R_4(\mu)}{R_3(\mu)} = E[U].
> $$
> Let $\ell(u)=\log g(u)$, then our plan is to show that that exists $\mu\in R$ and corresponding $C>0$ such that when $0\leq u\leq u_0:=C\left(\frac{M}{K}\right)^{1/3}$, $\frac{d \ell}{d u}(u)\ge 0$.
> To this end, let $\mu=1>0$ and $y=\frac{1}{u}+1$, then$$
>         \frac{d\ell}{du}(u)=\frac{1}{u}-2Ku+\frac{2M}{u^2y^3\left(e^{\frac{1}{y^2}}-1\right)}=\frac{1}{u}+2Ku\left(
>         \frac{M/K}{u^3y^3\left(e^{\frac{1}{y^2}}-1\right)}-1
>         \right).
>         $$
>         Since $1\leq y$, when $u\leq \left(\frac{M}{K}\right)^{\frac13}(e-1)^{-\frac13}-1$, we have
>         $$
>         \frac{M/K}{u^3y^3\left(e^{\frac{1}{y^2}}-1\right)}=\frac{M/K}{(u+1)^3\left(e^{\frac{1}{y^2}}-1\right)}\ge\frac{M/K}{(u+1)^3\left(e-1\right)}\ge 1.
>         $$
>         Take $C=(e-1)^{-\frac13}-2^{-\frac13}>0$ and then one can see that when $M\ge 2K$, $u_0:=C\left(\frac{M}{K}\right)^{1/3}\leq \left(\frac{M}{K}\right)^{\frac13}(e-1)^{-\frac13}-1$. Then when $0\leq u\leq u_0:=C\left(\frac{M}{K}\right)^{1/3}$, $\frac{d\ell}{du}(u)\ge 0$.
>
>
> **Question**:
> We appreciate your suggestion and will include this clarification. This choice is motivated by a sequence of prior studies on the FTPL algorithm in the multi-armed bandit (MAB) setting.
>
> Briefly, according to the decomposition in our Lemma 3.3, consider $X_1, \ldots, X_d$ as i.i.d. random variables drawn from the perturbation distribution $\mathcal{D}$. With a properly chosen learning rate, the penalty term corresponds to the expected sum of the top $m$ values among $\sqrt{n} X_1, \ldots, \sqrt{n} X_d$.
>
> Noting that the optimal regret bound in the adversarial setting is $\sqrt{nmd}$, the intuition is that we want to choose a distribution $\mathcal{D}$ such that the expectation of the sum of the top $m$ among $X_1, \ldots, X_d$ scales as $\sqrt{dm}$.
> According to our Lemma E.2, the Fréchet distribution with shape parameter $2$ satisfies this condition exactly. As for how we discovered that the Fréchet distribution satisfies such a condition, one can refer to classic results from extreme value theory.

---

> > ### Comment · Reviewer_7BAU · 2025-08-01
> >
> > Thank you for the effort and the concrete and solid rebuttal. I have reconsidered my evaluation and updated the rating.

---

> > > ### Author Response · Authors · 2025-08-01
> > > **Thank you!**
> > >
> > > We would like to express our sincere gratitude again for your constructive and valuable comments suggestions on our paper. We will incorporate the suggestions in the revised version.

---

### Note · Authors · 2025-08-12

We thank the reviewers for their constructive and meaningful discussions. Below we summarize and clarify the main outcomes of the exchange.

# Main Contributions

- We provide the first theoretical proof that Follow-the-Perturbed-Leader approaches Best-of-Both-Worlds for the $m$-set semi-bandit problem.

- We extend the standard analysis framework for Follow-the-Regularized-Leader algorithms to $m$-set semi-bandits — thereby simplifying Honda et al. [2023]’s proof.

- By establishing lower bounds, we demonstrate that our current approach has been pushed to its limit, and any further improvement would likely require adopting a different and more challenging line of analysis.

# Clarifications
We believe that we have addressed the vast majority of the reviewers’ concerns. In particular, reviewers pointed out the issue in our original Lemma 4.1. We resolved this by correspondingly modifying Lemma 4.2, and still proved that Follow-the-Perturbed-Leader approaches Best-of-Both-Worlds for the $m$-set semi-bandit problem. We provided the corresponding proof in the discussion, which has been acknowledged by the reviewers.

---

### Decision · Program_Chairs · 2025-09-17

**Decision:**

Accept (poster)

**Comment:**

This paper studies the m-set combinatorial semi-bandit problem and shows that FTPL with Fréchet perturbations achieves the near-optimal regret in the adversarial setting while also attaining logarithmic regret in the stochastic case, generalizing similar best-of-both-world guarantees of FTPL previously known for standard bandits (Honda et al. 2023). The authors further prove that the extra logarithmic factor in their bound is unavoidable without fundamentally new techniques.

Reviewers pointed out a mistake in the original proof, but were eventually convinced by the fix the authors proposed. Other than that, reviewers are generally positive about this paper. Given the long line of work on this best-of-both-world topic, I personally would be more excited if the analysis is significantly different from that of previous work and/or the results are presented for the general semi-bandit setting (instead of just m-set). That said, I would still recommend a borderline accept, given that it is a good addition to this line of work after all.